# Mesenchymal Stem/ Stromal Cells metabolomic and bioactive factors profiles: A comparative analysis on the umbilical cord and dental pulp derived Stem/ Stromal Cells secretome

Ana Rita Caseiro[1,2,3☯‡], Sílvia Santos Pedrosa[1,2☯‡], Galya Ivanova[4], Mariana Vieira Branquinho[1,2], André Almeida[2,5], Fátima Faria[6], Irina Amorim[6,7], Tiago Pereira[1,2], Ana Colette Maurício[1,2]*

1 Departamento de Clínicas Veterinárias, Instituto de Ciências Biomédicas de Abel Salazar (ICBAS), Universidade do Porto (UP), Rua de Jorge Viterbo Ferreira, Porto, Portugal, 2 Centro de Estudos de Ciência Animal (CECA), Instituto de Ciências, Tecnologias e Agroambiente da Universidade do Porto (ICETA), Rua D. Manuel II, Apartado, Porto, Portugal, 3 Escola Universitária Vasco da Gama (EUVG), Lordemão, Coimbra, Portugal, 4 REQUIMTE- LAQV, Departamento de Química e Bioquímica, Faculdade de Ciências, Universidade do Porto, Rua do Campo Alegre, Porto, Portugal, 5 Indústria Transformadora de Subprodutos —I.T.S, SA, Grupo ETSA, Rua Padre Adriano, Olivais do Machio, Santo Antão do Tojal, Loures, Portugal, 6 Departamento de Patologia e Imunologia Molecular, Instituto de Ciências Biomédicas de Abel Salazar (ICBAS), Universidade do Porto (UP), Rua de Jorge Viterbo Ferreira, Porto, Portugal, 7 i3S - Instituto de Investigação e Inovação da Universidade do Porto, Rua Alfredo Allen, Porto, Portugal

☯ These authors contributed equally to this work.
‡ These authors are first authors on this work.
* ana.colette@hotmail.com, acmauricio@icbas.up.pt

## Abstract

Mesenchymal Stem/ Stromal Cells assume a supporting role to the intrinsic mechanisms of tissue regeneration, a feature mostly assigned to the contents of their secretome. A comparative study on the metabolomic and bioactive molecules/factors content of the secretome of Mesenchymal Stem/ Stromal Cells derived from two expanding sources: the umbilical cord stroma and the dental pulp is presented and discussed. The metabolic profile (Nuclear Magnetic Resonance Spectroscopy) evidenced some differences in the metabolite dynamics through the conditioning period, particularly on the glucose metabolism. Despite, overall similar profiles are suggested. More prominent differences are highlighted for the bioactive factors (Multiplexing Laser Bear Analysis), in which Follistatin, Growth Regulates Protein, Hepatocyte Growth Factor, Interleukin-8 and Monocyte Chemotactic Protein-1 dominate in Umbilical Cord Mesenchymal Stem/ Stromal Cells secretion, while in Dental Pulp Stem/ Stromal Cells the Vascular Endothelial Growth Factor-A and Follistatin are more evident. The distinct secretory cocktail did not result in significantly different effects on endothelial cell populations dynamics including proliferation, migration, tube formation capacity and *in vivo* angiogenesis, or in chemotaxis for both Mesenchymal Stem/ Stromal Cells populations.

**Data Availability Statement:** All relevant data are within the manuscript and its Supporting Information files.

**Funding:** This research was supported by Programa Operacional Regional do Norte (ON.2 – O Novo Norte), QREN, FEDER with the project "iBone Therapies: Terapias inovadoras para a regeneração óssea", ref. NORTE-01-0247-FEDER-003262, and by the program COMPETE – Programa Operacional Factores de Competitividade, Projects PEst-OE/AGR/UI0211/2011 and PEst-C/EME/UI0285/2013 funding from FCT. This research was also supported by Programa Operacional Competitividade e Internacionalização (P2020), Fundos Europeus Estruturais e de Investimento (FEEI) and FCT with the project "BioMate – A novel bio-manufacturing system to produce bioactive scaffolds for tissue engineering" with reference PTDC/EMS-SIS/7032/2014 and by COMPETE 2020, from ANI – Projetos ID&T Empresas em Copromoção, Programas Operacionais POCI, by the project "insitu.Biomas - Reinvent biomanufacturing systems by using an usability approach for in situ clinic temporary implants fabrication" with the reference POCI-01-0247-FEDER-017771. This work received further financial support from the framework of QREN through Project NORTE-07-0124-FEDER-000066. The Bruker Avance III 600 HD spectrometer was purchased under the framework of QREN, through Project NORTE-07-0162-FEDER-000048 and is part of the Portuguese NMR Network created with support of FCT through Contract REDE/1517/RMN/2005, with funds from POCI 2010 (FEDER). Ana Rita Caseiro (SFRH/BD/101174/2014) acknowledges FCT, for financial support. The funders had no role in study design, data collection and analysis, decision to publish, or preparation of the manuscript.

**Competing interests:** The authors have declared that no competing interests exist.

# 1. Introduction

Mesenchymal Stem/ Stromal Cells (MSCs) are at the forefront of research for the development of cell-based therapies, due to their capacity to self-renew and differentiate into several cell types, to secrete soluble factors with paracrine actions, as well as due to their immunosuppressive and immunomodulatory properties [1–6].

MSCs have been described to reside on nearly every body tissue [7–9] since Friedenstein and colleagues firstly described the bone marrow derived population [10, 11].

Currently, the umbilical cord stroma (Whärton jelly) and the dental pulp may come to gain ground as sources for MSCs-based therapies, due to the non/ minimally invasive and ethically accepted collection procedures (umbilical cords and extracted healthy teeth were previously considered medical waste), as well as for the increasingly available private and public banking options worldwide [12].

The first evidence of the MSCs contribution to the healing processes was assigned to their specific differentiation skills, replacing the damaged native cells in their functions [13, 14]. However, current trends demonstrate that in some instances MSCs remain undifferentiated at the lesion site or in its vicinity, for limited periods of time, or even that only minimal percentages of the MSCs would effectively differentiate and integrate host tissues [15]. Regardless of their differentiation into tissue specific phenotypes, MSCs are often correlated to improved regenerative outcomes [16].

These observations were then attributed to the secretion products of those MSCs [17–19] and, in recent years, research has focused on deepening the knowledge on the effective composition of the MSCs secretion, in the form of soluble molecules or extracellular vesicles [6, 20–23].

In most tissues, the key for regenerative efficiency is the re-vascularization of the lesion site and MSCs have been associated with improved angiogenesis in a number of models of disease [24, 25]. As such, MSCs assume a supporting role to the intrinsic mechanisms of tissue regeneration, promoting the re-vascularization processes, providing adequate perfusion to active healing sites, as well as urging resident regenerative populations to home towards these locations [26].

Further, some groups investigated the extent to which the presence of the cells themselves was absolutely essential to the observation of beneficial effects, since regenerative benefit can be observed by the application of MSCs secretion products alone, conventionally designated as the secretome [8, 17–19, 27].

The secretome comprises a range of bioactive molecules/factors secreted to the extracellular space. Its composition is particular to individual cells and tissues, and is modulated in response to physiological and/or pathological stimuli [24]. The application of these cell-based products may bring several advantages to the advanced therapies field, namely the decreased cell number requirements and allocated cell storage necessities, ease of tailoring, quality control and dosing, reduced risk of rejection and malignancy, as well as the ready availability for administration in acute scenarios [28].

Therefore, studies on the composition of the MSCs secretome through metabolic analysis are a valuable tool to the comprehension of the underlying mechanisms to MSCs dynamics and therapeutic effects [29–32].

Metabolomic profiling techniques [33–37] yield information on targeted metabolites' structure and quantitative distribution [33, 34], and despite the significant progress made in the field of structural biology and bio-chemistry, the development and application of these techniques towards the MSCs secretome are still sparse [8].

We recently demonstrated the application of proton NMR spectroscopy and implementation of appropriate one (1D) and two (2D) dimensional NMR techniques to the analysis of the metabolic composition of Umbilical Cord Stem/ Stromal Cells (UC-MSCs) conditioned media and changes in the metabolic profile of the culture media in the process of conditioning [21].

Alongside the metabolite content, a wide range of growth factors, cytokines, chemokines and extracellular matrix components have already been identified in the CM obtained from differently sourced MSCs [38, 39], and many of them are known to impact on most tissues structure, function and regeneration [25, 40, 41]. Beyond modulating their surrounding environment MSCs are sensitive themselves to signaling factors, altering their secretory profile in response to, as an example, the presence of inflammatory cytokines. This evidence may contribute to the understanding of the eventual difference in the tissue response to cells or to their secretome alone [42], and to the development of strategies to manipulate secretion profiles towards specific needs.

In a recent study, we determined that UC-MSCs CM becomes rich in a range of proliferative and anti-apoptotic factors, particularly in transforming growth factor beta 1 (TGF-β1), epidermal growth factor (EGF), granulocyte and granulocyte-macrophage colony-stimulating factor (G-CSF and GM-CSF), platelet derived growth factor (PDGF-AA) and vascular endothelial growth factor (VEGF). Also, several other chemokines such as monocyte chemotactic protein-1 e 3 (MCP-1 and MCP-3), chemokine (C-C motif) ligand 5 (CCL5 or RANTES), GRO, and interleukin 8 (IL-8) were observed in increasing levels [21]. This previous analysis enabled the primary definition of UC-MSCs CM composition [21], which we herein expand and compare to Dental Pulp Stem/ Stromal Cells (DPSCs) CM profile.

The first part of the present study focuses on a comparative analysis of the metabolomic and bioactive factors secretion profile of UC-MSCs and DPSCs secretome, through the conditioning process. This aimed to characterize the metabolite profile of the collected CM and to enlighten on the metabolic pathways presiding to the process. Further, the bioactive factors content of the CM was analyzed, focusing on a series of growth factors, cytokines and chemokines, aiming at the comparison of the regenerative potential of the two MSCs population. Finally, the effects of the characterized CMs were addressed on *in vitro* and *in vivo* models of angiogenesis.

## 2. Materials and methods

### 2.1. Human Mesenchymal Stem/ Stromal Cells characterization and conditioning protocol

Umbilical Cord -MSCs [PromoCell™ (C-12971; Lot No. 1112304.2)] and DPSCs [AllCells, LLC (DP0037F, Lot No. DPSC090411-01)] were cultured with MEM-α, GlutaMAX™ (32561–029, Gibco®) supplemented with MSCs certified FBS (04-400-1A, Biological Industries Israel Beit-Haemek Ltd)(10% v/v), penicillin (100 U/mL)-streptomycin (100 μg/mL) (151140–122, Gibco®), and amphotericin B (2.5 μg/mL)(15290–026, Gibco®). Both cell types were maintained at 37 ˚C in a 5% $CO_2$ humidified atmosphere. Passage 4 cells were seeded in 75 $cm^3$ T-flasks until confluence was reached. Detachment of confluent cells was achieved by a 5 minutes incubation in 0,05% Trypsin-EDTA (Trypsin-EDTA 0.25%, 25200–072, Gibco®). Cells were re-suspended and counted using 0.4% Trypan Blue (T8154-20ML, Sigma-Aldrich®) and the Countess II FL Automated Cell Counter (Invitrogen™).

**2.1.1. Mesenchymal Stem/ Stromal Cells phenotype identity.** The surface marker profiles of UC-MSCs and DPSCs were confirmed though Flow Cytometry, with anti-positive (CD90, CD105, CD44) and negative marker (CD34, CD11b, CD19, CD45, MHC class II) antibodies (Human MSC Analysis Kit, 562245, BD Biosciences), as described in detail in [43].

Data was acquired using BD FACSCalibur™ 3 CA Becton Dickinson, BD Biosciences, and data was processed using FlowJo Engine X10.4 (v3.05478, LLC).

**2.1.2. Reverse transcriptase polymerase chain reaction.** Reverse transcriptase Polymerase chain reaction (RT-PCR) and qPCR targeting specific genes expressed by pluripotent stem cells was performed. Gene DNA sequences were downloaded from GenBank (www.ncbi.nlm.nih.gov/genbank) and aligned using the Clustal Omega bioinformatic tool from EMBL-EBI (http://www.ebi.ac.uk/Tools/msa/clustalo). The primers sequences are listed in **Table 1**.

Cultures were harvested and pelleted for total RNA extraction (High Pure RNA Isolation kit (Roche™)). DNA traces were eliminated with DNase I, and RNA quantity and quality assessed by Nanodrop ND-1000 Spectrophotometer, reading from 220 nm to 350 nm, and stored at -80˚C. After, cDNA was synthesized from the purified RNA (kit Ready-To-Go You-Prime First-Strand Beads (GE Healthcare®)).The synthesized cDNA, corresponding to the mRNA present in the sample, was run for the expression of six genes: two housekeeping genes (β-actin and GAPDH) and four genes used as pluripotent stem cells markers (c-kit, Oct-4, Nanog and ALP). Quantitative PCR (qPCR) was performed in a CFX96™ (BioRad®) apparatus using the iQ™ SYBR® Green Supermix (BioRad®). Each pair of primers targeting a gene was used to analyze its expression in the UC-MSCs and DPSCs cDNA, in duplicate, along with a negative control, through defined temperature cycles [95˚C for 4 minutes, 35 cycles comprising 95˚C for 20 seconds, 55˚C for 20 seconds and 72˚C for 30 seconds ending with Real-Time acquisition, and final extension of 72˚C for 7 minutes], and the number of cycle threshold for each well were recorded. The plate containing the amplified genes or qPCR products was kept in ice and observed in a 2% agarose gel (NuSieve® 3:1 Agarose (Lonza)) to check and reinforce the identity of the amplicons, in horizontal electrophoresis apparatus. Under 120 V potential difference for 40 minutes to separate the amplicons. Gel was then observed under UV light and pictures recorded using the GelDoc® 2000 (BioRad®) and Quantity One® software (BioRad®).

***2.1.3. Multilineage differentiation.*** Multilineage differentiation of the UC-MSCs and DPSCs was induced towards Osteogenic, Adipogenic and Chondrogenic phenotypes using specific differentiation media. Differentiation efficiency was assessed by Alizarin Red S, Oil Red O and Alcian Blue/ Sulfated Glycosaminoglycans (GAGs) quantification, as detailed in [43].

## 2.2. Metabolomic and bioactive factors secretion profiles

**2.2.1. Conditioned medium collection and analysis.** For the production of CM, both cellular populations were plated at 6000 cells/ cm$^2$, in triplicates, and cultured in standard culture medium (αMEM 10% FBS) until approximately 80% confluence was reached. Then, the cultures were washed to remove any trace of FBS and other supplements and placed in plain DMEM/F12 GlutaMAX™ (10565018, Gibco®) for 24 and 48 hours. Conditioned media samples were collected at both time-points, centrifuged 1800 x*g* for 10 min to remove any cellular debris and preserved at -80˚C until analysis.

**2.2.2 Metabolomic analysis.** Proton NMR spectroscopy was used to identify and quantify metabolites in plain/unconditioned and conditioned media collected at two time-points (24 and 48 hours) of UC-MSCs and DPSCs in proliferation. A total 600 μL aliquot of each sample was placed into 5 mm NMR tubes and 50 μL deuterium oxide (D$_2$O) containing 0.05 mM sodium trimethylsilyl-[2,2,3,3-d4]-propionate (TSP) was added as a chemical shift and quantitation standard. All NMR spectra were recorded at 300K on a Bruker Avance III 600 HD spectrometer, equipped with CryoProbe Prodigy. $^1$H NMR spectra with water suppression using a 1D NOESY (noesygppr1d) and a Carr-Purcell-Meiboom-Gill (CPMG, cpmgpr1d) pulse

**Table 1. List of primers used, target gene and size of the PCR product.**

| Primer | | GenBank target gene | PCR product |
|---|---|---|---|
| *Pluripotent Stem Cells Markers genes* | | | |
| **ALP human** | Fwd: 5'-CCTAAAAGGGCAGAAGAAGGAC-3' | NM_001632.4 | 444 bp |
| | Rev: 5'-TCCACCTAGGATCACGTCAATG-3' | | |
| **c-Kit human** | Fwd: 5'-AACGCTCGACTACCTGTGAA-3' | NM_000222 | 401 bp |
| | Rev: 5'-GACAGAATTGATCCGCACAG-3' | | |
| **Nanog human** | Fwd: 5'-CTTCCTCCATGGATCTGCTTATTC-3' | XM_011520851.1 | 265 bp |
| | Rev: 5'-AGGTCTTCACCTGTTTGTAGCTGAG-3' | | |
| **Oct4 human** | Fwd: 5'-GAAGCTGGAGAAGGAGAAGCT-3' | NM_002701.5 | 243 bp |
| | Rev: 5'-CAAGGGCCGCAGCTTACACAT-3' | | |
| *Housekeeping genes* | | | |
| **β-actin human** | Fwd: 5'-GGCACCCAGCACAATGAAGA-3' | NM_001101.3 | 100 bp |
| | Rev: 5'-CTGGAAGGTGGACAGCGAGGC-3' | | |
| **GAPDH human** | Fwd: 5'-AGCCGCATCTTCTTTTGCGTC-3' | NM_002046.5 | 815 bp |
| | Rev: 5'-TCATATTTGGCAGGTTTTTCT-3' | | |

sequences [44, 45] were acquired with spectral width of 10000 Hz, 32 K data points and 32 scans. The 1D NOESY spectra were collected using 5 s relaxation delay and mixing time of 100 ms. The CPMG experiments (cpmgpr1d) were acquired with relaxation delay 4 s, spin-echo delay between 0.4 and 0.6 ms, and a loop for T2 filter of 20 was used. All free induction decays (FIDs) were processed by 0.3 Hz line broadening and zero filling to 64 K, manually phased, and baseline corrected. Two dimensional 1H/1H COSY and TOCSY spectra were recorded in phase sensitive mode and with water suppression; a relaxation delay of 2 s, 16 or 32 scans, a total 2K data points in F2 and 256 or 512 data points in F1 over a spectral width of 10000 Hz. Two dimensional 1H/13C heteronuclear single quantum coherence (HSQC) experiments were carried out with a spectral width of 10000 Hz for 1H and 27000 Hz for 13C, relaxation delay 1,5 s, Fourier transform (FT) size 2K × 1K. The quantitative distribution of NMR-detectible metabolites in the samples was determined from the integral intensity of characteristic signals in $^1$H NMR spectra of the samples referenced to the integral intensity of TSP signal, considering the number of the contributing nuclei for that particular resonance signal [46].

**2.2.3. Bioactive factors detection and quantification.** Multiplexing LASER Bead Analysis was performed by Eve Technologies (Calgary, Alberta, Canada) using the Bio-Plex™ 200 system (Bio-Rad Laboratories, Inc., Hercules, CA, USA), similarly to previously described in [47]. The multiplexing technology is based on color-coded polystyrene beads with unique color / fluorophore signatures, and a dual-laser system and a flow-cytometry system. One laser activates the fluorescent dye within the beads which identifies the specific analyte, the second laser excites the fluorescent conjugate (streptavidin-phycoerythrin), and the amount of the conjugate detected by the analyzer is in direct proportion to the amount of the target analyte. The results are quantified according to a standard curve.

A broad panel of 57 bioactive factor was assayed [Discovery Assay® TGFβ 3-Plex Cytokine Array (Eve Technologies Corp); Milliplex Human Angiogenesis / Growth Factor kit Discovery Assay® and the Human Cytokine Array / Chemokine Array (Millipore, St. Charles, MO, USA)], including angiopoietin-2 (Ang), bone morphogenetic protein 9 (BMP-9), EGF, endoglin (ENG), endothelin-1 (EDN1), eotaxin-1, fibroblast growth factor 1 and 2 (FGF-1 and -2), fms-related tyrosine kinase 3 ligand (Flt-3L), follistatin (Fst), fractalkine, G-CSF, GM-CSF, GRO(pan), heparin-binding-EGF (HB-EGF), hepatocyte growth factor (HGF), interferon-alpha 2 (IFNα2), interferon-gama (IFNγ), several interleukins (IL-1α, IL-1β, IL-1ra, IL-2, IL-3,

IL-4, IL-5, IL-6, IL-7, IL-8, IL-9, IL-10, IL-12 (p40), IL-12 (p70), IL-13, IL-15, IL-17A, IL-18), interferon gama-induced protein 10 (IP-10), leptin, MCP-1, MCP-3, macrophage-derived chemokine (MDC), macrophage inflammatory protein-1 alpha (MIP-1α), macrophage inflammatory protein-1 beta (MIP-1β), platelet-derived growth factor-AA (PDGF-AA) and -AB/BB (PDGF-AB/BB), placental growth factor (PLGF), RANTES/ CCL5, soluble CD40 ligand (sCD40L), transforming growth factor alpha (TGFα), transforming growth beta 1, 2 and 3 (TGF-β 1, -2, and -3), tumor necrosis factor alpha (TNFα), tumor necrosis factor beta (TNFβ), VEGF-A, -C, and -D.

## 2.3. Effects of MSCs CM on angiogenesis

**2.3.1. Conditioned medium concentration protocol.** Conditioned Medium obtained from the UC-MSCs and DPSCs was concentrated 5 times (5x), using Pierce™ Protein Concentrator, 3k MWCO, 5–20 mL (88525, Thermo Scientific). Tubes were sterilized *as per* manufacturer's instructions, through immersion in 70% (v/v) ethanol for 30 minutes, followed by 2 cycles of 70% ethanol and DPBS (14190144, Gibco®) centrifugations at 3000 x*g*, for 10 minutes each). Tubes were air dried for 10 minutes and CM placed ate the top compartment. Samples were centrifuged until 5x concentration of sample volume in the upper compartment was achieved.

**2.3.2. Human umbilical vein endothelial cells expansion and maintenance.** Human Umbilical Vein Endothelial Cells (UVECs) were obtained from Sigma® (200P-05N; Lot No. 3257, Cell Applications, Inc) and expanded using specific expansion medium (211–500, Cell Applications, Inc). Cells were maintained at 37˚C and 95% humidified atmosphere with 5% $CO_2$ environment. Passage 4–5 cultures were utilized in the presented assays.

**2.3.3. Cell viability assay.** Cells were seeded in 24 well plate at 6000 cells/cm$^2$. The cells were allowed to adhere in UVECs expansion medium, overnight. After this period, culture media was replaced by CM supplemented culture media [composed of 80% UVECs expansion medium and 20% 5x concentrated UC-MSCs' CM or DPSCs' CM; control medium was composed of 80% UVECs expansion media and 20% DPBS].

At every time point (0, 24, 48, 72 and 96 hours), the culture media was removed and fresh adequate culture media was added to each well, with 10% (v/v) of 10x PrestoBlue® cell viability reagent (A13262, Thermo Fisher Scientific, Molecular Probes, Invitrogen), and UVECs were incubated for 1 hour at 37˚C, 5% $CO_2$. The absorbance of the cell culture supernatant was read at 570 nm and 595 nm in a Thermo Scientific Multiskan FC plate reader, to assess for changes in cell viability. Absorbance readings were normalized, and data corrected to unseeded control wells' readings.

**2.3.4. Senescence and apoptosis assays.** To assess for signs of senescence in the cellular populations, β-Galactosidase activity was assayed. Cells were plated in 96-well plates, similarly as described for the cell viability assessment. After 48 hours in CM-supplemented and control media, culture media was removed, and cells were washed once with DPBS and incubated with β-Galactosidase Assay Reagent (75705, Thermo Fisher Scientific), for 30 minutes at 37˚C. Absorbance was read at 405 nm.

Cultured UVECs were further assessed for apoptosis events, through the detection of Annexin V and Propidium Iodine (PI) staining (BMS500FI, eBioscience). For such, cells cultured for 2 days in each condition were harvested using Trypsin-EDTA, counted and resuspended in provided Binding Buffer. Cells were incubated with Annexin V-FITC and PI, *as per* manufacturer's protocol, and FACS analysis was performed using a Coulter Epics XL Flow Cytometer (Beckman Coulter Inc., Miami, FL, USA). Flow cytometry data was processed using FlowJo Engine X10.4 (v3.05478, LLC).

**2.3.5. Migration assays.** Human UVECs, UC-MSCs and DPSCs were plated at 10000 cells/cm$^2$ in a 12 well-plate. Cells were further allowed to expand up to 90% confluency and the centre of the well scratched using a P200 sterile pipette tip.

Wells were individually photographed, and standard culture medium was replaced by defined experimental conditions and appropriate controls [UVECs assay—CM media 80% UVECs expansion medium and 20% 5x concentrated UC-MSCs' CM or DPSCs' CM; Control Medium: 80% UVECs expansion media and 20% DPBS; Complete Medium: 100% UVECs expansion media; MSCs assay—CM media 80% αMEM 10% FBS and 20% 5x concentrated UC-MSCs' CM or DPSCs' CM; Control Medium: 80% αMEM 10% FBS and 20% DPBS; Complete Medium: 100% αMEM 10% FBS].

Cells were cultured in the conditions above described and observed for migration and proliferation into the scratched area. Photographs were taken at 0, 10, 14 and 16 hours (for UVECs) and 0, 6 and 24 hours (for MSCs) after experimental media addition and media was changed after each procedure. Photographs were obtained from marked areas along each scratch line, allowing for the monitoring of cell response in multiple areas (Axiovert 40 CFL, Zeiss®).

Photographs were then analysed using the ImageJ Software (ImageJ 1.51k, NIH, USA) and the scratched area was measured in time sequenced images, using the 'MRI Wound Healing Tool' (http://dev.mri.cnrs.fr/projects/imagej-macros/wiki/Wound_Healing_Tool). Six measurements per condition were considered and the results were presented as 'percentage decrease in scratched area'.

**2.3.6. *In vitro* endothelial tube formation assay.** For the evaluation of the effect of the CMs on the *in vitro* endothelial tube formation capacity of the UVECs, cells were plated on Matrigel® (Growth Factor Reduced Matrigel®, Cat. 354230, Corning®) coated 96-well culture plates. A final density of 16000 cells *per* well were plated and specific media added to each group. Endothelial tubes formation was accompanied for up to 12 hours, and photographic record at 40x magnification was obtained at this timepoint (Axiovert 40 CFL, Zeiss®). Recorded images were then analysed using the ImageJ Software (ImageJ 1.51k, NIH, USA).

Recorded measurements concerned covered area (%), branching points (#), total tubes (#) and total tube length (px), total loops (#), total loop area (px$^2$) and total loop perimeter (px).

**2.3.7. *In vivo* vascularization assay.** For the *in vivo* assessment, adult male Sasco Sprague Dawley rats [300-350g body weight (b.w.)] were selected. Experimental animals were housed in environmentally controlled facilities, under 12h light-dark cycles. Standard rodent chow and water were provided *ad libitum*. Normal cage activities were allowed, under standard laboratory conditions. All experimental procedures were approved by the Organism Responsible for Animal Welfare (ORBEA) of the Abel Salazar Institute for Biomedical Sciences (ICBAS), University of Porto (UP) (project 165/2016) and by the Veterinary Authorities of Portugal (DGAV) (project DGAV: 2018-07-11 014510), complying with Directive 2010/63/EU of the European Parliament and the European Communities Council Directive 86/609/EEC. Humane Endpoints were considered as recommended by the OECD Guidance Document on the Recognition, Assessment and Use of Clinical Signs as Humane Endpoints for Experimental Animals Used in Safety Evaluation (2000). Before surgical procedure, animals were maintained for two weeks under normal routine, for acclimation.

Anaesthesia protocol consisted on intraperitoneal (IP) injection of xylazine-ketamine mixture (Rompun® 20 mg/mL, Bayer®, 12 mg/kg b.w., and Imalgene 1000®, Merial®, 100 mg/kg b.w.). The dorsum of each animal was clipped, and the skin asepsis performed with iodopovidone 10% solution (Betadine®).

Matrigel® was thoroughly mixed with the Control, Complete and Conditioned Media (80% Matrigel® and 20% of the respective medium). The formulation was maintained in ice,

to prevent Matrigel® gelation. The dorsum was virtually divided into four quadrants (each assigned to an experimental group) and a final volume of 500 μL was injected subcutaneously. Upon injection, the formation of a protruding plug was confirmed. Animals were recovered from anaesthesia. After 7 days, animals were placed again under general anaesthesia and sacrificed by lethal intraperitoneal injection of 5% sodium pentobarbital (Eutasil® 200 mg/mL, Ceva).

Injection sites were confirmed and dorsal subcutaneous tissue enclosing the vascularised plug were collected and preserved in paraformaldehyde (3.7–4% buffered to pH 7, Panreac AppliChem®).

Collected tissues were processed for routine histopathologic analysis. Sequential sections (3 μm) were prepared and stained with haematoxylin-eosin (H&E) and assessed for the presence and pattern of endotheliocyte/ capillary penetration into the matrix plug. For the immunohistochemical study, sections were deparaffinized in xylene and rehydrated in sequential graded alcohols. Antigen retrieval was performed in EDTA, pH 8,0 for 30 minutes in water bath 100º C. The NovolinkTM Max-Polymer detection system (Novocastra, Newcastle, UK) was used for visualization, according to the manufacturer's instructions. Slides were then incubated with anti-VEGFR2 antibody (clone 55B11; Cell Signaling Technology, Boston, USA), diluted 1:300, overnight at 4ºC. Colour was developed with 3.3- diamino-benzidine (DAB; Sigma, St. Louis, MO, USA) and sections were then counterstained with haematoxylin, dehydrated and mounted. Sections of normal rat skin were used as positive control and negative controls were performed by replacing the primary antibody with another of the same immunoglobulin isotype.

The tissue areas containing Matrigel® plugs and the surrounding tissue were microscopically evaluated in order to identify the regions of highest vascular density. Vessels which showed unequivocal brown VEGFR2 immunostaining were counted manually at higher magnification (×40) in at least 10 different regions of each sample.

## 2.4. Statistical analysis

Data was plotted and treated using GraphPad Prism version 6.00 (GraphPad Software, La Jolla California USA). Data was further analyzed for significant differences. Multiple comparison tests were performed by one-way ANOVA supplemented with Tukey's HSD post-hoc test. Differences were considered statistically significant at $P < 0.05$. Significance of the results is indicated according to P values with one, two, three or four of the symbols (*) corresponding to $0.01 \leq P < 0.05$; $0.001 \leq P < 0.01$; $0.0001 \leq P < 0.001$ and $P < 0.0001$, respectively.

## 3. Results

### 3.1. Mesenchymal Stem/ Stromal Cells characterization and conditioning protocol

Mesenchymal Stem/ Stromal Cells populations depicted characteristic phenotypical markers (Fig 1), presenting $\geq$ 92% positive population for CD90, CD105 and CD44, and $\leq$ 2% negative marking for CD34, CD11b, CD19, CD45 and MHC II. Exception was noted for DPSCs population with regard to CD90, which marked positive for only 87–91% of assessed population (Positive population: UC-MSCs—CD90+: 99.23 ± 0.12; CD105+: 99.07 ± 0.09; CD44+: 95.70 ± 1.65; NEG+*: 0.13 ± 0.08; DPSCs: CD90+: 90.50 ± 0.35; CD105+: 99.30 ± 0.06; CD44+: 99.80 ± 0.00; NEG+*: 0.13 ± 0.04; *NEG+ cocktail corresponds to anti-CD34; -CD11b; -CD19; -CD45; and -MHC II antibodies).

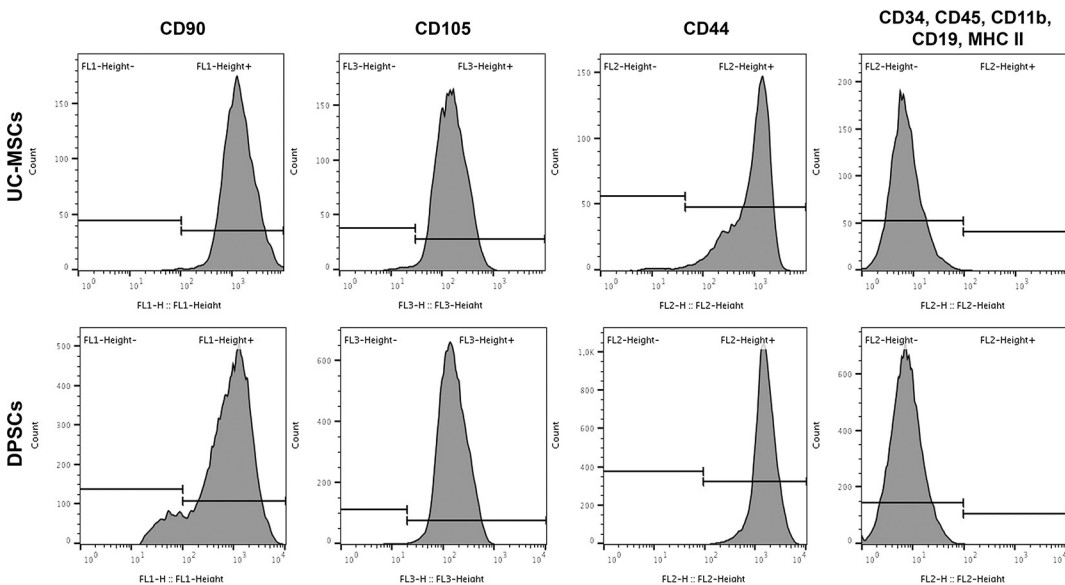

**Fig 1. Surface marker expression for MSCs' identity of UC-MSCs and DPSCs.** Positivity for CD90, CD105 and CD44, negativity for a cocktail including CD34, CD11b, CD19, CD45 and MHC II, assessed by Flow cytometry.

RNA was purified from both UC-MSCs and DPSCs and converted to cDNA using adequate procedures. Primers targeting two housekeeping genes (β-actin and GAPDH) and four typical pluripotency markers (c-kit, Oct-4, Nanog and ALP) were used to support its identity. **Table 2** details on the average of Threshold cycle (Ct) values. Total RNA was successfully purified from UC-MSCs and DPSCs using the High Pure RNA Isolation kit (Roche), obtaining a concentration of 883.2 ng/μl from UC-MSCs and 158.7 ng/μl from DPSCs. Volumes were adjusted to use only 1.5 μg of total RNA to synthetize corresponding cDNA. This cDNA was used as template in the qPCR reaction. The Ct values show a reasonable amplification of the target genes, in both cell types, resulting in an active expression of these genes in both UC-MSCs and DPSCs. The agarose gel confirms the identity of the genes from the observed molecular weight.

The strongest expression is observed in the housekeeping genes, while the lowest expression was observed for the Oct-4 sequence (as evidenced by the lowers and higher ΔCt values determined for the amplification cycles, respectively) (**Table 2**).

The differentiation capacity of MSCs towards three mesodermal lineages was confirmed (**Fig 2**). Macroscopic observation revealed calcified matrix formation, evidenced by ARS staining in both UC-MSCs and DPSCs under osteodifferentiation conditions. Semi-quantitative

**Table 2. Average of Delta Threshold cycle (Ct) values.** Of the amplification of the selected genes from UC-MSCs and DPSCs.

| Gene | ΔCt value | | | | | |
|---|---|---|---|---|---|---|
| | UC-MSCs | | | DPSCs | | |
| *Pluripotent Stem Cells Markers genes* | | | | | | |
| **ALP** | 6.03 | ± | 0.03 | 6.36 | ± | 0.01 |
| **c-kit** | 1.96 | ± | 0.01 | 1.49 | ± | 0.02 |
| **Nanog** | 6.41 | ± | 0.06 | 6.15 | ± | 0.02 |
| **Oct-4** | 7.95 | ± | 0.00 | 7.26 | ± | 0.01 |

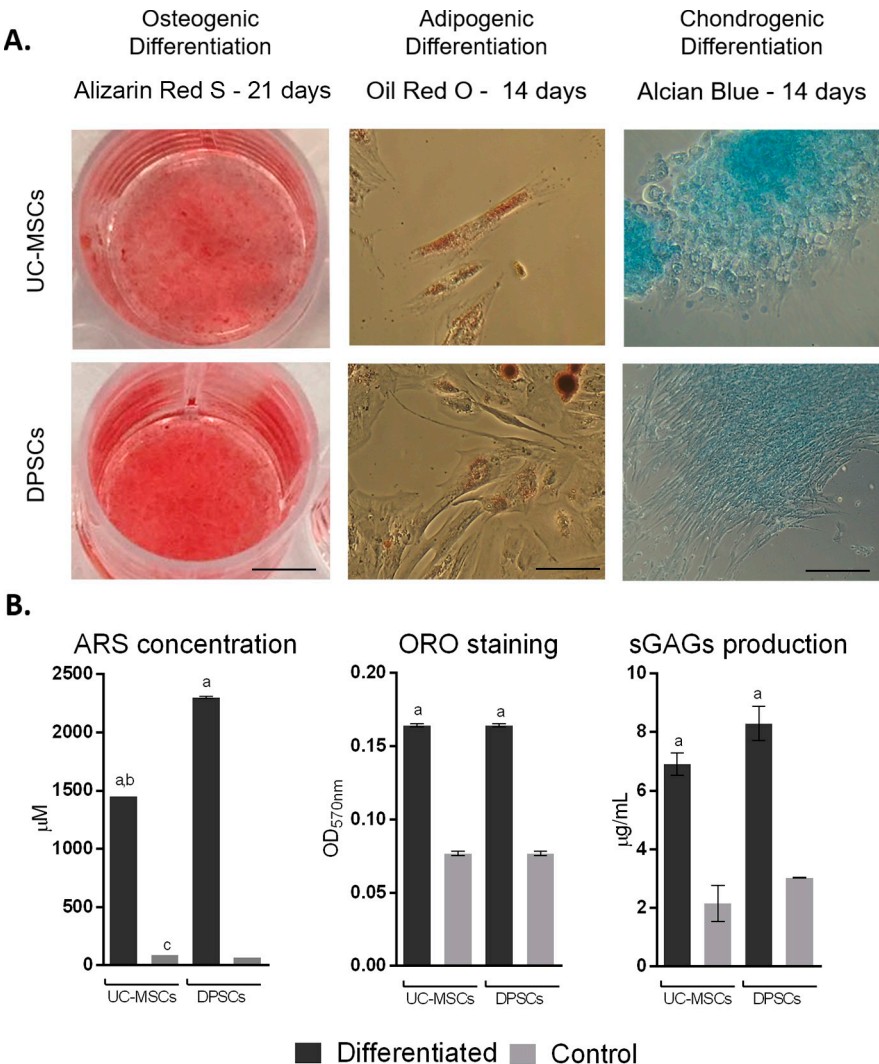

**Fig 2. Multilineage differentiation.** A) Qualitative evaluation—Osteogenic differentiation: Alizarin Red S (ARS) staining after 21 days (scale bar = 6000 μm); Adipogenic differentiation: Oil Red O (ORO) staining after 14 days (scale bar = 100 μm); Chondrogenic differentiation: Alcian blue staining after 14 days (scale bar = 400 μm). B) Semi-Quantitative evaluation—Osteogenic differentiation: ARS concentration (μM) after 21 days; Adipogenic differentiation: Oil Red O ($OD_{570nm}$) after 14 days; and C) Chondrogenic differentiation: Sulfated GAGs production (μg/ml) after 14 days, assessed by Blyscan™ Glycosaminoglycan Assay (Biocolor, UK). Control: Undifferentiated control; Results presented as Mean ± SEM. a: significantly different from undifferentiated group; b: UC-MSCs differentiated group significantly different from DPSCs differentiated group; c: UC-MSCs undifferentiated group significantly different from DPSCs undifferentiated group ($P<0.05$).

analysis confirmed observed differentiation, further indicating an increased efficiency and osteogenic propensity of DPSCs when compared to UC-MSCs (2299.31 ± 7.35 μM and 1449.57 ± 0.00 μM of ARS, respectively). In adipogenic differentiation, microscopic observation of ORO stained cells evidenced comparable lipid droplets accumulation in differentiated UC-MSCs and DPSCs. Spectrophotometric quantification of ORO staining confirmed observed differentiation and indicate no significant difference between UC-MSCs and DPSCs efficiency. Chondrogenic differentiation groups presented marked light blue staining under macroscopic and microscopic observation, indicating proteoglycan deposition by Alcian Blue staining, confirmed by sGAGs production quantification. UC-MSCs and DPSCs presented no

difference in terms of sGAGs production within the differentiated and the undifferentiated wells.

## 3.2. Metabolomic and bioactive factors secretion profiles

### 3.2.1. Conditioned medium collection and analysis

Populations from both tissue sources were successfully seeded and reached the desired density within 3 to 4 days. For conditioning time, cells remained well attached and preserved their characteristic spindled shape, although a decrease in the proliferation rate could be empirically observed.

### 3.2.2. Metabolomic analysis

Proton NMR spectroscopy was used to define the metabolic profiles of UC-MSCs and DPSCs through the analysis of produced CMs. To identify metabolite changes induced by DPSCs and UC-MSCs conditioning, various 1D and 2D NMR experiments were performed, and $^1$H NMR spectra acquired using appropriate NMR techniques for suppression of the solvent and other interfering signals. The same experimental conditions (temperature, sample/solvent volume, standard (TSP) concentration, NMR acquisition and processing parameters) were used in order to avoid external interferences and to identify particular metabolite variations. The assignment of the proton resonances in NMR spectra and the assessment of the metabolite composition of the samples were achieved by high resolution $^1$H NMR spectra analysis, considering specific NMR parameters which reflect structural characteristics of the respective species. The results were verified by the implementation of appropriate 2D NMR ($^1$H/$^1$H COSY, $^1$H/$^1$H TOCSY, $^1$H/$^{13}$C HSQC) spectroscopic techniques and compared to available data in the literature [33, 34]. The assignment of the resonance signals of metabolites in $^1$H NMR spectra of the samples studied is presented in **Table 3**. Characteristic resonance signals of metabolites identified in $^1$H NMR spectra of Plain (unconditioned), UC-MSCs and DPSCs CM are labelled in **Fig 3**.

The visual inspection of the $^1$H NMR spectra (**Fig 3**) suggests similar profile shapes but different concentration distribution of the metabolites. To assess characteristic metabolic changes

**Table 3. $^1$H NMR chemical shifts and multiplicity.** Of the main metabolites observed in the spectra of Plain, UC-MSCs and DPSCs CM.

| Metabolites | Abbrev. | Chemical Shifts (ppm) multiplicity |
|---|---|---|
| Acetate | Ace | 1.92(s, CH$_3$) |
| Alanine | Ala | 1.48(d, bCH$_3$), 3.78(q, aCH) |
| Choline | Cho | 3.19 (s, CH$_3$), 3.51(dd, CH$_2$), 4.06(dd, CH$_2$), |
| Ethanol | Et | 1.19 (t, CH$_3$), 3.57 (q, CH$_2$) |
| Formate | For | 8.46 (CH) |
| Glutamate | Glu | 2.36(t, gCH$_2$), 2.08(m, bCH$_2$), 3.74(dd, aCH) |
| Glutamine | Gln | 2.44(t, gCH$_2$), 2.51/2.04(m, bCH$_2$), 3.76(dd, aCH) |
| Lactate | Lac | 1.34(d, bCH$_3$), 4.13(q, aCH) |
| Nicotinamide | NA | 7.58 (dd, CH), 8.24 (dd, CH), 8.70 (dd, CH), 8.94 (s, CH) |
| Pyruvate | CH$_3$ | 2.39(s) |
| Tyrosine | Tyr | 3.05/3.18 (bCH$_2$), 3.93 (aCH), 7.19(d, H2,6), 6.90(d, H3,5) |
| α-Glucose | α–Γλυ | 5.24(d, H1), 3.54(dd, H2), 3.72(t, H3), 3.41(t, H4), 3.47(ddd, H5), 3.91/3.72 (dd, H6) |
| β-Glucose | β–Γλυ | 4.65(d, H1), 3.24(t, H2), 3.49(t, H3), 3.42(t, H4), 3.84(m, H5,6), 3.78(dd, H6) |

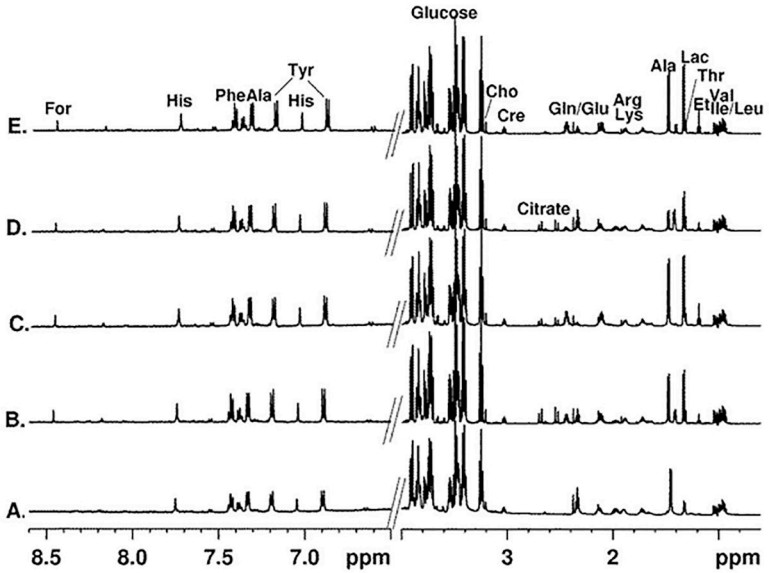

**Fig 3. Average 600 MHz [1]H NMR spectra.** (in $H_2O$ with 10% $D_2O$) of Plain Medium (A), UC-MSCs CM collected after 24 (B) and 48 hours (C), and of DPSCs CM collected after 24 (D) and 48 hours (E). The intensity in the aromatic spectral area (6.5–8.6 ppm) incremented by 10x; the water signal is excluded from the spectra.

occurring during hMSCs conditioning, comparative quantitative analysis using [1]H NMR spectra of the samples from the three groups was performed.

The metabolites concentration was determined from the integral intensity of characteristic signals in [1]H NMR spectra of the samples referenced to the integral intensity of TSP signal used as an internal standard and considering the number of the contributing nuclei for that particular resonance signal. The concentration of the metabolites identified in the [1]H NMR

**Table 4. Metabolite concentration (μM).** Calculated from [1]H-NMR spectra of unconditioned (Plain Medium), and UC-MSCs and DPSCs conditioned media, after 24 and 48 hours. Data presented in Mean ± SD.

| Metabolite | Group | Chem. Shift (ppm) | Concentration (μM) | | | | | | | | | | | | | | | | |
|---|---|---|---|---|---|---|---|---|---|---|---|---|---|---|---|---|---|---|---|
| | | | Plain Medium | | | UC-MSCs 24h | | | UC-MSCs 48h | | | DPSCs 24h | | | DPSCs 48h | | |
| Acetate | $CH_3$ | 1.93 | 2.91 | ± | 0.02 | 6.16 | ± | 0.81 | 5.15 | ± | 0.34 | 2.03 | ± | 1.17 | 2.00 | ± | 0.06 |
| Alanine | $CH_3$ | 1.48 | 16.34 | ± | 0.52 | 120.65 | ± | 1.76 | 165.76 | ± | 2.75 | 46.42 | ± | 0.89 | 141.99 | ± | 6.17 |
| Choline | $N(CH_3)_3$ | 3.21 | 2.53 | ± | 0.09 | 4.60 | ± | 0.89 | 3.87 | ± | 0.17 | 3.93 | ± | 0.14 | 4.20 | ± | 0.52 |
| Ethanol | $CH_3$ | 1.19 | 0.84 | ± | 0.13 | 8.22 | ± | 0.18 | 16.13 | ± | 0.07 | 8.47 | ± | 0.56 | 21.00 | ± | 0.30 |
| Formate | CH | 8.46 | 0.39 | ± | 0.19 | 1.58 | ± | 0.27 | 1.92 | ± | 0.77 | 1.07 | ± | 0.09 | 1.48 | ± | 0.26 |
| GlutaMAX I (L-Ala) | $CH_3$ | 1.42 | 94.64 | ± | 1.24 | 77.09 | ± | 0.18 | 4.97 | ± | 1.94 | 88.83 | ± | 1.77 | 25.39 | ± | 9.74 |
| GlutaMAX II (L-Glu) | $CH_2$ | 2.34 | 153.38 | ± | 4.29 | 160.99 | ± | 4.52 | 29.72 | ± | 0.36 | 160.17 | ± | 5.22 | 62.61 | ± | 8.03 |
| Glutamine | $CH_2$ | 2.45 | 6.62 | ± | 0.44 | 134.98 | ± | 4.32 | 175.09 | ± | 2.54 | 37.85 | ± | 0.60 | 157.38 | ± | 15.51 |
| Lactate | CH | 4.12 | 11.36 | ± | 0.44 | 111.44 | ± | 8.04 | 131.75 | ± | 6.58 | 87.20 | ± | 1.45 | 137.52 | ± | 11.95 |
| Nicotinamide | CH | 8.94 | 0.44 | ± | 0.37 | 2.55 | ± | 0.72 | 2.04 | ± | 1.77 | 1.14 | ± | 0.13 | 1.16 | ± | 0.13 |
| Pyruvate | $CH_3$ | 2.37 | 14.40 | ± | 0.15 | 15.67 | ± | 1.07 | 7.53 | ± | 0.37 | 12.49 | ± | 4.69 | 6.97 | ± | 0.22 |
| Tyrosine | 2CH | 7.19 | 21.59 | ± | 0.26 | 35.59 | ± | 1.07 | 27.51 | ± | 0.44 | 26.61 | ± | 2.11 | 27.36 | ± | 1.39 |
| α-Glucose | CH | 5.24 | 773.63 | ± | 1.52 | 1 112.15 | ± | 58.57 | 813.21 | ± | 16.77 | 805.17 | ± | 25.38 | 860.69 | ± | 15.68 |
| β-Glucose | CH | 4.65 | 945.19 | ± | 14.00 | 1 307.92 | ± | 9.58 | 1 038.61 | ± | 63.04 | 913.16 | ± | 2.84 | 998.30 | ± | 19.05 |

spectra of the samples was calculated using the equation [48]:

$$Wx = \frac{Ix \times Nstd \times Mx \times mstd}{Istd \times Nx \times Mstd}$$

Where:

- **Wx** represent the mass of the metabolite;

- **Mx** and **Mstd** are the molar masses of the metabolite and the standard (TSP was used), respectively;

- **Ix** and **Istd** are the integrated signal area of the metabolite and the standard, respectively;

- **Nx** and **Nstd** are the number of protons in the integrated signal area of the metabolite and the standard;

- **mStd** represent the mass of the standard.

The concentration of the metabolites (μM) identified in each group, as well as the chemical shift and assignment of the signal used for quantification are presented in **Table 4**. Statistical significance of encountered differences in Metabolite concentration (μM) is presented in **S1 Table**.

All $^1$H NMR spectra are dominated by the proton resonance signals of glucose but several low molecular weight compounds such as amino acids (e.g. alanine, tyrosine), organic acids (e.g. lactate, acetate, citrate, formate) and choline are identified (**Fig 3**).

Additionally, in the spectra of the Plain Medium, resonance signals characteristic for L-ala-nyl-L-glutamine (GlutaMAX$^{TM}$ I (L-Ala) at 1.42 and GlutaMAX$^{TM}$ II (L-Glu) at 2.37 ppm) dipeptide from GlutaMAX$^{TM}$ are clearly observed. The intensity of these resonances significantly decreases through the cell conditioning process, which is especially evident after 48 hours of culture conditioning. Glutamine and alanine, the resulting compound from Gluta-MAX$^{TM}$ and glutamate's metabolism, expectedly rise through culturing times. Remarkably, UC-MSCs' CM increase in these metabolites' content occurs at 24 hours, before significant GlutaMAX$^{TM}$ consumption can be observed. For the DPSCs population, this process appears to occur at a slower rate (**Fig 4A** and **Fig 4B**).

Glucose is the most prominent metabolite identified in the spectra and therefore in greatest concentration. In the media collected after 24 hours of conditioning of UC-MSCs, a significant increase of α and β-Glucose content is observed, when compared to the Plain Medium (which was supplied to the culture at the start of the conditioning period), as well as when compared to the DPSCs at the same conditioning time. This initial spike in glucose content decreases in the following hours, reaching back to values not significantly different from the Plain Medium. Human DPSCs present no significant variation in glucose content throughout the study period.

Pyruvate is an energetic compound that is catabolised through aerobic or anaerobic pathways. The analysis of the Plan Media confirms it is one of the supplemented metabolites. Both cell types appear to present a tendency for pyruvate consumption in the 48 hours of conditioning (**Fig 4C** and **Fig 4D**), but significance in such reduction is only found for UC-MSCs. Catabolites resulting from its metabolism are also detected in the spectra, such as acetate, lactate and ethanol. Acetate accumulated in UC-MSCs cultures, particularly in the first 24h (**Fig 4C**). In DPSCs CM, Acetate concentrations remained stable though the evaluation (**Fig 4D**). Lactate, on the other hand, accumulated strongly within the first 24 hours in both cases, with little increase afterwards, reaching maximum concentration at 48 hours of incubation. Ethanol

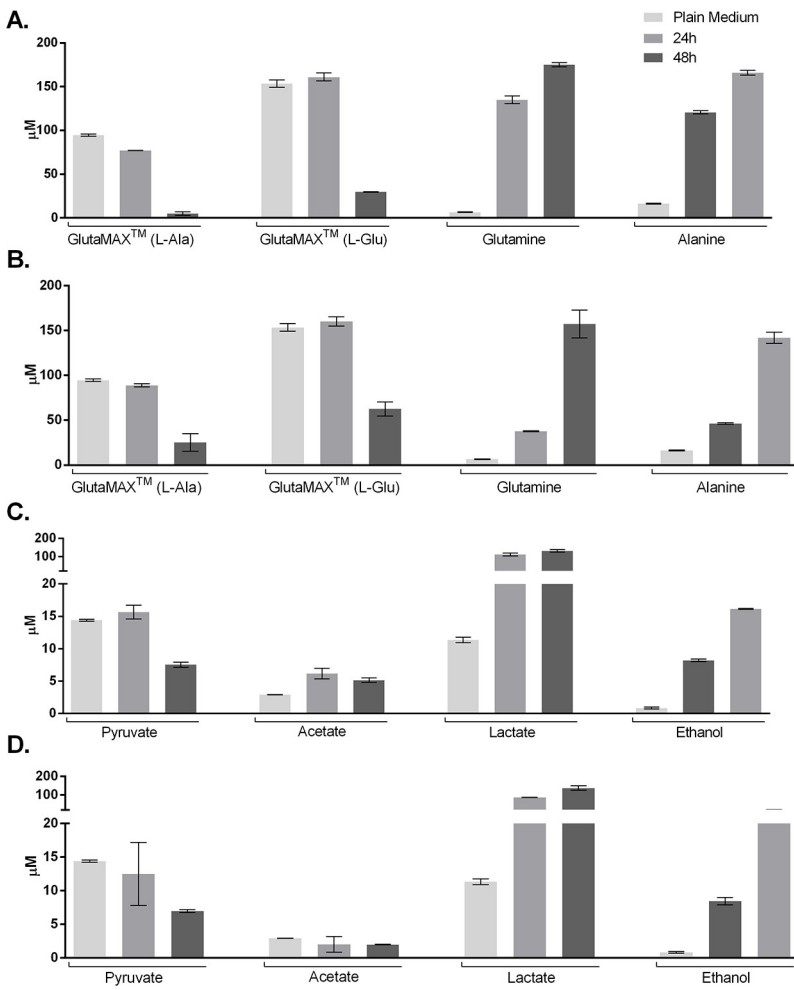

**Fig 4. Metabolite dynamics.** Dynamics of GlutaMAX[TM] breakdown to glutamine and alanine of UC-MSCs (A) and DPSCs (B) populations; and Dynamics of Pyruvate catabolism into acetate, lactate and ethanol of UC-MSCs (C) and DPSCs (D) populations. Statistical significance of observed differences available in **S1 Table**, as Supporting Information.

presented a steady rate of accumulation in the culture supernatant for both UC-MSCs and DPSCs.

Other compounds such as choline, formate, tyrosine and nicotinamide are also detected at low levels in both Plain and Conditioned media.

**3.2.3. Bioactive factors detection and quantification.** Multiplexing LASER Bead Analysis was performed for the detection of a total of 57 Bioactive factors in the CM obtained from UC-MSCs and DPSCs, after 24 and 48 hours of conditioning. Obtained measurements were normalized to the Unconditioned/ Plain medium (naturally devoid of artificial bioactive factors), run as blank sample. Fluorescent readings were plotted against a standard curve and calculated concentrations are presented in **Table 5**. Statistical significance of encountered differences in the detected Bioactive factors (pg/mL) is presented in **S2 Table**.

After 24 hours of conditioning most factors presented no or little expression. Some of the factors gained expression after 48 hours, becoming detectable and, occasionally, in significant concentrations. Overall, none to minimal signal intensity was detected in 36 of the 57 assayed factors, combining the two timepoints.

**Table 5. Bioactive factors detection and quantification (pg/mL).** Calculated from Multiplexing LASER Bead Analysis of UC-MSCs and DPSCs CM after 24 and 48 hours of conditioning. Data presented in Mean ± SEM. Bioactive factors presenting no to minimal signal intensity in grey. ND, not detected.

| Bioactive Factor | Concentration (pg/mL) | | | | | | | | | | | |
|---|---|---|---|---|---|---|---|---|---|---|---|---|
| | UC-MSCs 24h | | | UC-MSCs 48h | | | DPSCs 24h | | | DPSCs 48h | | |
| Angiop-2 | ND | | | 10.66 | ± | 0.45 | ND | | | 1.86 | ± | 1.00 |
| BMP-9 | ND | | | 0.22 | ± | 0.02 | ND | | | 0.00 | ± | 0.00 |
| EGF | 0.23 | ± | 0.13 | 0.07 | ± | 0.04 | 0.28 | ± | 0.02 | 0.68 | ± | 0.10 |
| Endoglin | ND | | | 47.74 | ± | 3.25 | 0.93 | ± | 0.93 | 2.56 | ± | 0.39 |
| Endothelin-1 | ND | | | 1.21 | ± | 0.07 | ND | | | 0.03 | ± | 0.03 |
| Eotaxin-1 | ND | | | 1.50 | ± | 0.49 | 4.37 | ± | 2.08 | 3.09 | ± | 1.54 |
| FGF-1 | ND | | | 1.97 | ± | 0.63 | ND | | | 0.26 | ± | 0.22 |
| FGF-2 | ND | | | 212.27 | ± | 12.39 | ND | | | 7.06 | ± | 7.06 |
| Flt-3L | ND | | | 1.37 | ± | 0.69 | ND | | | 0.76 | ± | 0.22 |
| Follistatin | 143.45 | ± | 10.69 | 1101.53 | ± | 293.83 | 202.04 | ± | 29.56 | 1309.40 | ± | 110.60 |
| Fractalkine | 3.37 | ± | 0.95 | 23.76 | ± | 13.03 | ND | | | ND | | |
| G-CSF | ND | | | 1041.30 | ± | 113.67 | ND | | | 0.49 | ± | 0.25 |
| GM-CSF | 1.24 | ± | 0.34 | 2.45 | ± | 1.07 | ND | | | 1.04 | ± | 0.25 |
| GRO pan | 17.80 | ± | 7.49 | 966.84 | ± | 177.44 | ND | | | 12.19 | ± | 1.45 |
| HB-EGF | 0.02 | ± | 0.00 | 0.97 | ± | 0.11 | 0.04 | ± | 0.04 | 0.10 | ± | 0.03 |
| HGF | 113.99 | ± | 7.25 | 4436.31 | ± | 516.74 | 0.37 | ± | 0.37 | 69.86 | ± | 18.50 |
| IFNα2 | 3.19 | ± | 0.99 | 3.04 | ± | 0.41 | 10.95 | ± | 5.08 | 2.28 | ± | 1.11 |
| IFNγ | 0.27 | ± | 0.05 | 1.22 | ± | 0.11 | 0.40 | ± | 0.16 | ND | | |
| IL-10 | ND | | | 0.29 | ± | 0.05 | ND | | | ND | | |
| IL-12(p40) | 2.80 | ± | 0.89 | 3.49 | ± | 0.63 | 0.64 | ± | 0.11 | ND | | |
| IL-12(p70) | 0.32 | ± | 0.17 | 0.40 | ± | 0.04 | 0.38 | ± | 0.03 | 0.41 | ± | 0.14 |
| IL-13 | 1.02 | ± | 0.06 | 0.76 | ± | 0.12 | 0.27 | ± | 0.04 | 1.15 | ± | 0.13 |
| IL-15 | 0.65 | ± | 0.02 | 0.80 | ± | 0.19 | 0.37 | ± | 0.04 | 0.49 | ± | 0.03 |
| IL-17A | ND | | | ND | | | ND | | | ND | | |
| IL-18 | ND | | | ND | | | ND | | | ND | | |
| IL-1B | 0.56 | ± | 0.08 | 0.35 | ± | 0.05 | 0.50 | ± | 0.26 | 0.43 | ± | 0.17 |
| IL-1RA | 0.54 | ± | 0.14 | 0.78 | ± | 0.27 | ND | | | ND | | |
| IL-1α | 4.45 | ± | 0.31 | 7.10 | ± | 0.73 | ND | | | ND | | |
| IL-2 | ND | | | ND | | | ND | | | ND | | |
| IL-3 | 0.38 | ± | 0.21 | ND | | | ND | | | ND | | |
| IL-4 | 0.81 | ± | 0.35 | ND | | | ND | | | ND | | |
| IL-5 | ND | | | ND | | | 0.01 | ± | 0.00 | 0.05 | ± | 0.03 |
| IL-6 | 31.24 | ± | 2.47 | 470.56 | ± | 33.61 | ND | | | ND | | |
| IL-7 | 0.12 | ± | 0.05 | 0.65 | ± | 0.10 | ND | | | ND | | |
| IL-8 | 48.18 | ± | 2.89 | 1863.05 | ± | 10.77 | 0.02 | ± | 0.01 | 1.23 | ± | 0.14 |
| IL-9 | ND | | | 0.30 | ± | 0.06 | ND | | | 0.38 | ± | 0.29 |
| IP-10 | 1.90 | ± | 0.10 | ND | | | ND | | | 1.23 | ± | 0.10 |
| Leptin | 1.76 | ± | 0.00 | 216.56 | ± | 17.66 | 39.62 | ± | 24.47 | 12.09 | ± | 12.09 |
| MCP-1 | 972.36 | ± | 83.42 | 2626.08 | ± | 236.18 | N.D. | | | 0.44 | ± | 0.33 |
| MCP-3 | 6.84 | ± | 3.45 | 53.53 | ± | 6.48 | 2.43 | ± | 0.89 | 7.23 | ± | 2.77 |
| MDC | ND | | | 32,13 | ± | 6,16 | ND | | | ND | | |
| MIP-1α | ND | | | ND | | | ND | | | ND | | |
| MIP-1β | ND | | | ND | | | ND | | | ND | | |
| PDGF-AA | 0.14 | ± | 0.01 | 0.56 | ± | 0.04 | ND | | | 0,02 | ± | 0,01 |
| PDGF-BB | 2.06 | ± | 0.56 | 3.15 | ± | 1.35 | 2.24 | ± | 0.67 | 4.29 | ± | 1.97 |

*(Continued)*

**Table 5.** (Continued)

| Bioactive Factor | Concentration (pg/mL) | | | | | | | | | | | |
|---|---|---|---|---|---|---|---|---|---|---|---|---|
| | UC-MSCs 24h | | | UC-MSCs 48h | | | DPSCs 24h | | | DPSCs 48h | | |
| PLGF | 0.17 | ± | 0.04 | 0.11 | ± | 0.03 | 0.13 | ± | 0.06 | 4.35 | ± | 0.30 |
| RANTES | 12.35 | ± | 2.32 | 73.61 | ± | 5.56 | 4.52 | ± | 1.58 | 1.53 | ± | 1.12 |
| sCD4L | ND | | | ND | | | ND | | | ND | | |
| TGF-α | ND | | | ND | | | 0.26 | ± | 0.08 | ND | | |
| TGF-β1 | 11.96 | ± | 2.18 | 582.49 | ± | 56.37 | 4.08 | ± | 2.08 | 53.01 | ± | 23.04 |
| TGF-β2 | 0.69 | ± | 0.15 | 174.20 | ± | 17.52 | 0.13 | ± | 0.13 | 29.33 | ± | 5.09 |
| TGF-β3 | 0.09 | ± | 0.09 | 1.77 | ± | 1.25 | ND | | | 0.20 | ± | 0.16 |
| TNFα | 0.05 | ± | 0.01 | 0.24 | ± | 0.07 | ND | | | 0.08 | ± | 0.00 |
| TNFβ | ND | | | 0.86 | ± | 0.17 | ND | | | 1.38 | ± | 0.27 |
| VEGF-A | 17.25 | ± | 3.48 | 1.34 | ± | 0.28 | 14.31 | ± | 8.75 | 2041.55 | ± | 45.66 |
| VEGF-C | 1.90 | ± | 0.13 | 486.48 | ± | 23.48 | ND | | | 30.71 | ± | 1.67 |
| VEGF-D | 0.57 | ± | 0.22 | 1.90 | ± | 0.48 | 0.52 | ± | 0.17 | ND | | |

A similar pattern of superior secretion by UC-MSCs is observed throughout, highlighting HGF, IL-8, and MCP-1 expressive quantities. Exception is appointed to follistatin and, most significantly, to VEGF-A, where DPSCs depict increased levels when compared to UC-MSCs.

## 3.3. Effects of MSCs CM on angiogenesis

### 3.3.1. Cell viability, senescence and apoptosis assays.
The CM obtained from MSCs cultures was employed as culture medium supplement for *in vitro* culture of UVECs, to assess for its ability to sustain the activity and proliferation of the endothelial populations. PrestoBlue® viability was used as a metabolic indicator. Obtained corrected absorbance values are

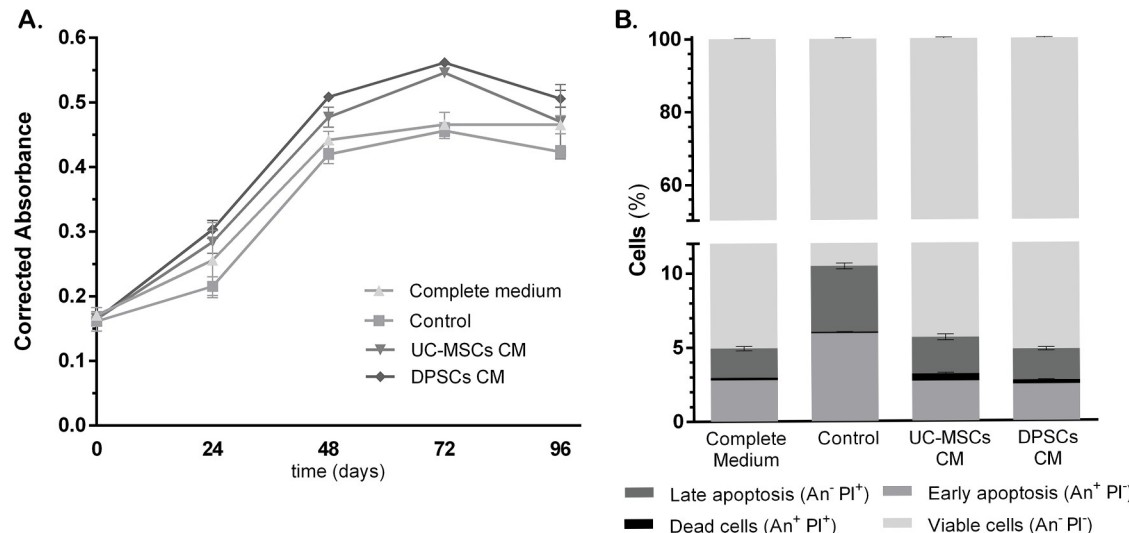

**Fig 5. Corrected absorbance readings of Presto Blue® Assay of UVECs.** At 0, 24, 48, 72 and 96 hours of expansion in control or CMs supplemented media. Values are presented as Mean ± SD (A). Statistical significance of observed differences available in S3 Table, as Supporting Information; Apoptosis (Annexin-V/ PI) assay of UVECs, after 48 hours of expansion in Complete, Control or CMs supplemented media. Results presented as percentage (%) of cells of Viable, Early Apoptotic, Late Apoptotic and Death cells, as Mean ± SD (B). Statistical significance of observed differences available in **S4 Table**, as Supporting Information.

**Table 6. Corrected absorbance readings of Presto Blue® Assay of UVECs.** At 0, 24, 48, 72 and 96 hours of expansion in control or CMs supplemented media. Values are presented as Mean ± SD.

| Corrected Absorbance | Complete Medium | | | Control | | | UC-MSCs CM | | | DPSCs CM | | |
|---|---|---|---|---|---|---|---|---|---|---|---|---|
| 0 h | 0.171 | ± | 0.012 | 0.161 | ± | 0.015 | 0.165 | ± | 0.011 | 0.165 | ± | 0.008 |
| 24 h | 0.254 | ± | 0.056 | 0.215 | ± | 0.014 | 0.284 | ± | 0.018 | 0.304 | ± | 0.014 |
| 48 h | 0.442 | ± | 0.014 | 0.420 | ± | 0.015 | 0.477 | ± | 0.015 | 0.509 | ± | 0.007 |
| 72 h | 0.464 | ± | 0.015 | 0.456 | ± | 0.012 | 0.546 | ± | 0.005 | 0.562 | ± | 0.003 |
| 96 h | 0.465 | ± | 0.014 | 0.424 | ± | 0.010 | 0.459 | ± | 0.040 | 0.502 | ± | 0.019 |

presented in **Fig 5** and **Table 6** (statistical differences on observed measurements detailed in **S3 Table**).

At 24 hours, UVECs cultured in medium supplemented with DPSCs CM depict slightly increased activity, with no other noteworthy differences. As time progresses (48 and 72 hours), both DPCSs and UC-MSCs CM supplemented groups sustain superior cellular activity than Complete and Control media. At about 72 hours of culture, the UVECs cultured in CM supplemented media reached confluence and entered the plateau stage. The activity of Control medium group also decreased from that point onwards, although cellular confluence was unmatched to the CM supplemented groups. At 96 hours, only the Complete medium continued to present an increase in cellular metabolism but remained slightly inferior to that provided by DPCSs CM.

Senescent events on the cultured populations were assessed through the β-galactosidade activity. Control medium presented increased enzyme activity when compared do the CM supplemented groups (Corrected absorbance, as mean ± SD–Control: 0.0025±0.0030; UC-MSCs CM 0.0003±0.0004; DPCSs CM: 0.0000±0.0000; $0.001 \leq P < 0.01$). CM supplemented groups presented no significant difference from the UVECs maintained in their standard Complete medium (Corrected absorbance, as mean ± SD: 0.0009±0.0007).

As for apoptosis events in the cultured population, none of the CM supplemented groups revealed noteworthy impact in cellular viability (**Fig 5B, Table 7** and **S4 Table**). The most striking observation is that the Control group presented increased events of early and late stage apoptosis, significantly decreasing the percentage of viable cells. As for the groups supplemented with DPSCs or UC-MSCs CM, the first presented increased cellular viability, as opposed to the early apoptotic and cell death events identified in the UC-MSCs CM supplemented group. The CM supplemented groups matched the events of each stage recorded for UVECs cultured in their standard Complete media, except for the early apoptosis, where DPSCs CM provided an inferior percentage.

**Table 7. Apoptosis (Annexin-V/ PI) assay of UVECs.** After 48 hours of expansion in Complete, Control or CMs supplemented media. Results presented as percentage (%) of cells of Viable, Early Apoptotic, Late Apoptotic and Dead cells, as Mean ± SD.

| Apoptosis (Annexin V/ PI) | Complete Medium | | | Control | | | UC-MSCs CM | | | DPSCs CM | | |
|---|---|---|---|---|---|---|---|---|---|---|---|---|
| Viable Cells | 95.07 | ± | 0.25 | 89.47 | ± | 0.41 | 94.33 | ± | 0.41 | 95.17 | ± | 0.29 |
| Early Apoptosis | 2.79 | ± | 0.06 | 5.93 | ± | 0.09 | 2.71 | ± | 0.06 | 2.47 | ± | 0.05 |
| Late Apoptosis | 1.97 | ± | 0.21 | 4.43 | ± | 0.28 | 2.71 | ± | 0.28 | 2.09 | ± | 0.16 |
| Dead Cells | 0.17 | ± | 0.01 | 0.11 | ± | 0.02 | 0.48 | ± | 0.10 | 0.27 | ± | 0.07 |

### 3.3.2. Migration assays

After wound induction on the cellular monolayers with the pipette tip, the area of the produced defect did not differ between groups/wells, deeming them adequate for comparison through the healing time.

In one instance, we investigated the chemotactic effect of the MSCs CMs on endothelial populations, represented by HUVECs. No significant differences in recovered area of the wound were identified until 14 hours into the assay, when DPSCs CM supplemented groups depicted increased percentage of covered wound area, when compared to the Complete and Control media ($0.0001{\leq}P{<}0.001$) (**Fig 6** and **Table 8**). At the final assessment time (16 hours), both CM supplemented groups presented over 90% area coverage, while Complete and Control media groups remained under 85% coverage of the originally induced 'scratch' ($0.0001{\leq}P{<}0.001$ to $P{<}0.0001$).

Further, we assessed the effect of the CMs on the chemotaxis of the two MSCs populations under study. After 6 hours of 'scratching' and media replacement of the MSCs monolayers, no

**UVECs Migration Assay**

| | Complete Medium | Control | UC-MSCs CM | DPSCs CM |
|---|---|---|---|---|
| **0h** | | | | |
| | | 0% | | |
| **10h** | | | | |
| | 59,9 % | 53,8 % | 59,7 % | 62,9 % |
| **14h** | | | | |
| | 67,8 % | 68,6 % | 77,2 % | 87,0 % |
| **16h** | | | | |
| | 84,0 % | 84,6 % | 90,1 % | 90,3 % |

**Fig 6. Migration assay of UVECs.** At 0, 10, 14 and 16 hours in control or CMs supplemented media. Values are presented as percentage (%) of wound area coverage from the 0 hours baseline (scale bar = 200 μm).

**Table 8. Migration assay of UVECs.** At 0, 10, 14 and 16 hours in control or CMs supplemented media, wounded area in $10^5$ pixels. Values are presented as Mean ± SD.

| Area ($10^5$ pixels) | Complete Medium | | | Control | | | UC-MSCs CM | | | DPSCs CM | | |
|---|---|---|---|---|---|---|---|---|---|---|---|---|
| 0 h | 33.207 | ± | 2.198 | 33.378 | ± | 1.573 | 35.089 | ± | 0.718 | 33.486 | ± | 2.322 |
| 10 h | 13.332 | ± | 1.547 | 15.436 | ± | 2.372 | 14.132 | ± | 2.694 | 12.431 | ± | 3.408 |
| 14 h | 10.703 | ± | 3.403 | 10.490 | ± | 2.028 | 7.993 | ± | 3.641 | 4.366 | ± | 2.235 |
| 16 h | 5.298 | ± | 1.202 | 5.132 | ± | 0.915 | 3.470 | ± | 0.295 | 3.244 | ± | 0.779 |

differences were observed in cellular migration in Control and Complete Medium groups between, and within the two MSCs sources (**Fig 7** and **Table 9**). The CM supplemented groups depicted superior area, with exception for CMs supplemented DPSCs, which did not differ significantly from the same population cultured in αMEM 10% FBS. No differences were patent in UC-MSCS and DPSCs supplemented with either CM ($0.0001 \leq P < 0.001$ to $P < 0.0001$).

At the final assessment of 24 hours, UC-MSCs in CM supplemented media presented increased migration into the area than the un-supplemented Control and αMEM 10% FBS groups. They also presented increased motility than the DPSCs populations ($0.0001 \leq P < 0.001$ to $P < 0.0001$). No difference in response to the two CMs was observed to within each MSCs type, but UC-MSCs responded with superior migration when paired to the DPSCs ($0.0001 \leq P < 0.001$ to $P < 0.0001$).

**3.3.3. In vitro endothelial tube formation assay.** The *in vitro* angiogenic potential of the CMs was evaluated through the tube formation capacity of UVECs, cultured in Complete, Control, UC-MSCs and DPSCs CM supplemented media. Control group presented the least effective tube formation capacity, while CM supplemented groups provided enhanced *in vitro* angiogenic potential (**Fig 8**).

It was noticeable that UVECs exposed to UC-MSCs and DPSCs CM supplemented media formed consistently well-defined tubular networks, with evenly distributed and regularly shaped loops in the Matrigel® matrix. Complete medium supplementation resulted in defined tubes, although loop distribution appears less homogeneous than in the CMs groups. Occasional events of incomplete tube connection from the branching points were observed. In the Control group, the dominant observation is the incomplete bridging attempts and short unconnected tubes, resulting in large and hill shaped loops.

Regarding the number of branching points, no differences were encountered between groups. In terms of total loops (formed from the connection of the branched tubes) and total tube length, Complete medium and CM supplemented groups showed significative superior performance, when compared to Control medium. Although both CMs provided increased results over the Complete medium group, statistical significance was only noted for the DPSCs CM supplemented group. No significant difference was found between the two CMs. The surface area covered by UVECs reflected the tube formation efficiency observed (**Fig 8** and **Table 10**).

**3.3.4. In vivo vascularization assay.** After 7 days of subcutaneous implantation, the Matrigel® plugs were evidenced and collected. Discrete vascular penetration was observed macroscopically in all groups.

Microscopically, the Complete Medium and the UC-MSCs CM groups presented apparent increased capillary density than the Control group (**Fig 9**). In the CM groups, the capillary penetration reached deeper into the matrix. These observations were further confirmed through VEGFR2 staining of the microvessels within the subcutaneous tissue at the penetrating interface with the Matrigel® plug (**Fig 9** and **Table 11**).

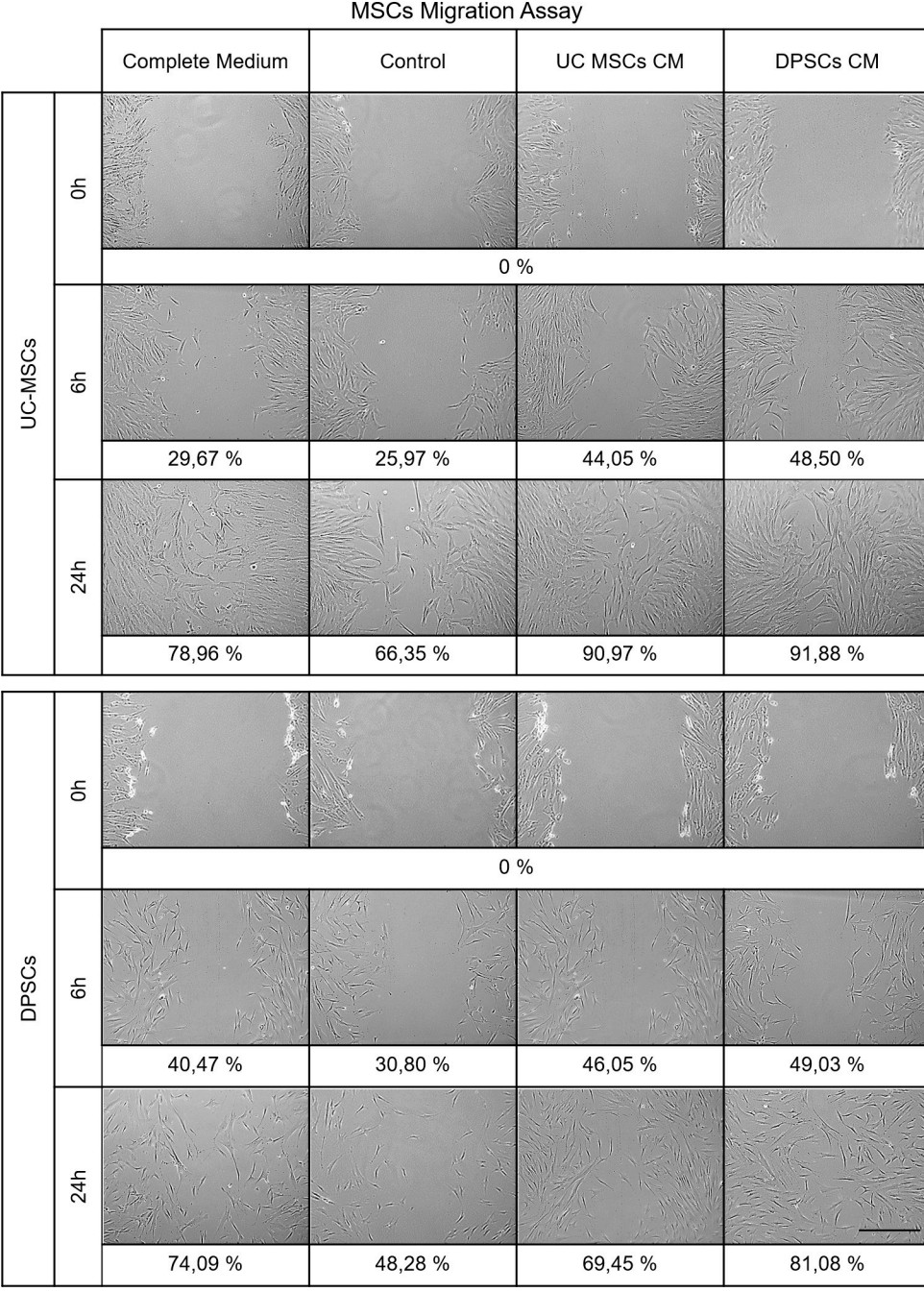

**Fig 7. Migration assay of hMSCs.** At 0, 6 and 24 hours of expansion in control or CMs supplemented media. Values are presented as percentage (%) of wound area coverage from the 0 hours baseline (scale bar = 200 μm).

## 4. Discussion and conclusions

The disclosure of the composition of MSCs populations' secretome is key to the comprehension of the underlying mechanisms of their therapeutic action. Therefore, it is key to understand the basal profile of secretion of these MSCs populations, to grant basis for the selection of the MSCs systems most suitable for each intended therapeutic application. For such, we

**Table 9. Migration assay of hMSCs.** At 0, 6 and 24 hours of expansion in control or CMs supplemented media, wounded area in $10^5$ pixels. Values are presented as Mean ± SD.

| Area (10⁵ pixels) | | Complete Medium | | | Control | | | UC-MSCs CM | | | DPSCs CM | | |
|---|---|---|---|---|---|---|---|---|---|---|---|---|---|
| UC-MSCs | 0 h | 41.42 | ± | 2.61 | 37.55 | ± | 4.54 | 35.06 | ± | 3.98 | 36.56 | ± | 5.14 |
| | 6 h | 29.13 | ± | 3.82 | 27.80 | ± | 2.73 | 19.62 | ± | 2.98 | 18.83 | ± | 2.11 |
| | 24 h | 8.72 | ± | 3.45 | 12.64 | ± | 2.43 | 3.17 | ± | 1.08 | 2.97 | ± | 1.97 |
| DPSCs | 0 h | 42.51 | ± | 5.87 | 41.26 | ± | 3.27 | 41.42 | ± | 2.87 | 42.74 | ± | 2.05 |
| | 6 h | 25.30 | ± | 3.71 | 28.55 | ± | 3.53 | 22.34 | ± | 2.28 | 21.79 | ± | 2.75 |
| | 24 h | 11.01 | ± | 3.65 | 21.34 | ± | 5.73 | 12.65 | ± | 2.54 | 8.09 | ± | 2.66 |

conducted an analytical study focusing on the investigation of the metabolomic and bioactive factors composition of the secretome of hMSCs originated from two of the most promising sources for medical applications: the UC-MSCs and the DPSCs. These sources may come to gain ground for MSCs-based therapies due to the non-/ minimally invasive and ethically accepted collection procedures, as well as for the increasingly available private and public banking options worldwide.

The international scientific community has long debated the criteria for MSCs character assignment and issued a conciliatory guiding line of features for the definition of populations classified as MSCs [2], considering their phenotypic and differentiation capacities, which was confirmed on the hMSCs herein explored. We observed an increased capacity for DPSCs to differentiate towards mineralising populations, as reported by [49], but did not confirm their decreased capacity for adipogenesis since both UC-MSCs and DPSCs displayed comparable Oil Red O uptake. Further, a panel of specific genes has been reported to correlate with the proliferative capacity of undifferentiated MSCs and pluripotency/ multilineage differentiation capacity. Both UC-MSCs and DPSCs presented comparable expression of ALP, c-kit and Nanog. ALP has been described to relate to the proliferative stage of undifferentiated stem cells of embryonic and mesenchymal origin, while Nanog expression in MSCs is associated with the transition from *in vivo* quiescence to adaptation to *in vitro* growth condition. Both genes are described to downregulate as lineage commitment occurs, except for the osteoblastic phenotype conversion, where it is characteristically upregulated [50, 51]. C-kit is the gene coding for the receptor for the stem cell factor, also observed highly proliferative populations and related to differential lineage differentiation capacity [52]. Oct-4 was expressed to a lower extent in both cellular populations. Oct-4, along with Nanog, is associated to MSCs populations plasticity [53]. Other authors report that Oct-4 was not identified in cultured human adult MSCs [51].

These populations were therefore expanded in culture and elicited to secrete bioactive molecules to the surrounding media, through the conditioning process. The conditioning process was handled under serum-free conditions, so that the only factors detected in the CM were those produced by the UC-MSCs and DPSCs and not those bared to the system by exogenous supplementation.

Addressing the obtained CM from a metabolomic perspective (through $^1$H-NMR spectroscopy), we identified glucose as the dominant metabolite in the spectra. Nuschke et al. recently investigated the metabolism of glucose in MSCs populations (bone marrow) and determined this to be a major limiting factor for MSCs survival. They describe a steady consumption rate for glucose in semi- and confluent cultures [54], a tendency that we could not observe in the cultured UC-MSCs and DPSCs. Indeed, UC-MSCs present an increase in both α- and β-glucose content, suggesting a possible engagement to gluconeogenesis and the utilization of other

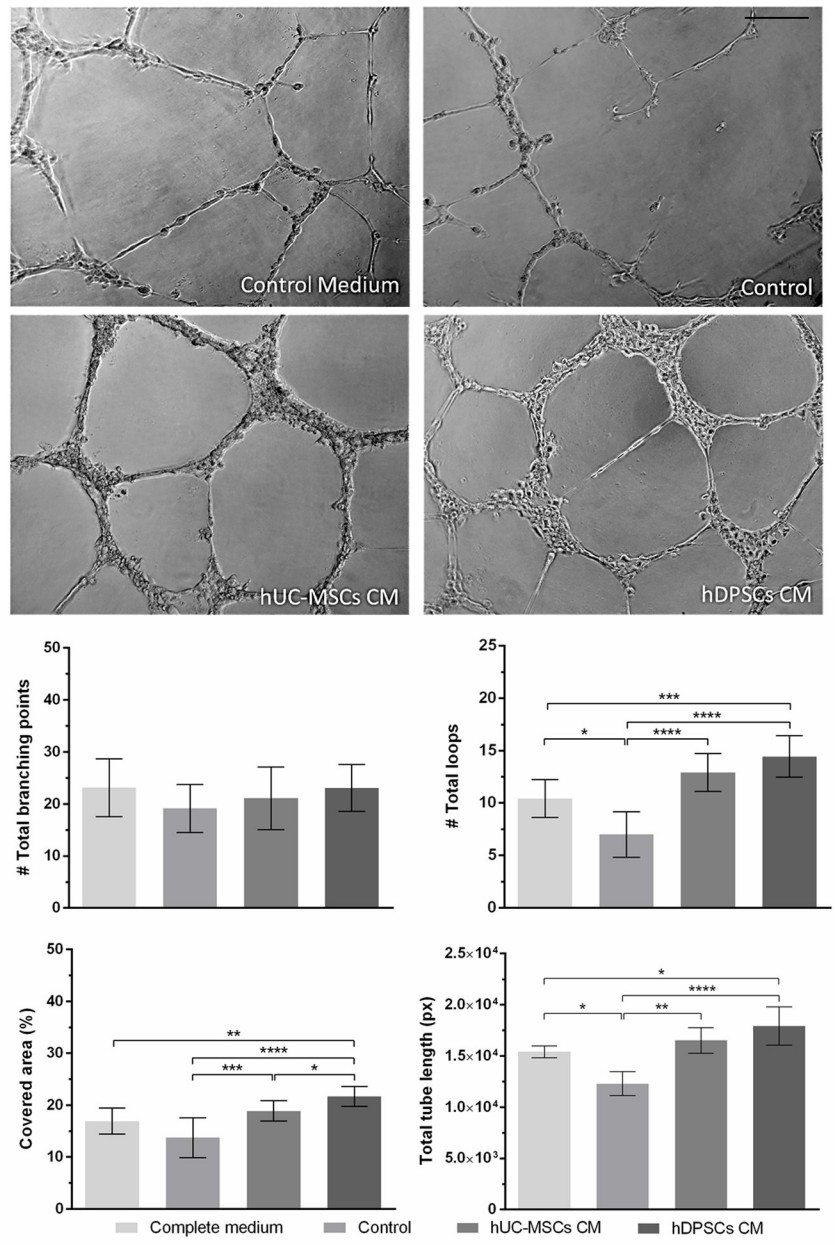

**Fig 8. Tube formation assay.** After 12 hours of UVECs exposure to Complete, Control, UC-MSCs or DPSCs supplemented media (upper panel). Graphical representation of total branching points, total loops, covered area and total tube length observed in each group (lower panel) (scale bar = 200 μm). Values are presented as Mean ± SD. Significant differences indicated according to P values with one, two, three or four of the symbols (*) corresponding to $0.01 \leq P < 0.05$; $0.001 \leq P < 0.01$; $0.0001 \leq P < 0.001$ and $P < 0.0001$, respectively.

bioenergetic substrates for energy production. The observed profiles suggest that, although significant amounts of glucose are provided by plain media, conditioning cells do not solely rely on glycolytic pathways for ATP and NADP production.

Pyruvate is therefore suggested as one of the crucial energetic metabolites that enters a variety of metabolic pathways, for the aerobic production of ATP [through its conversion into acetate/acetylCoA and entrance into the citric acid cycle]. Pyruvate is detected at relatively low

**Table 10. Tube formation assay.** After 12 hours of UVECs exposure to Complete, Control, UC-MSCs or DPSCs supplemented media: total branching points, total loops, covered area and total tube length observed in each group. Values are presented as Mean ± SD.

| Tube Formation Assay | Complete medium | | | Control | | | UC-MSCs CM | | | DPSCs CM | | |
|---|---|---|---|---|---|---|---|---|---|---|---|---|
| Covered area (%) | 16.93 | ± | 2.50 | 13.75 | ± | 3.82 | 18.89 | ± | 1.98 | 21.69 | ± | 1.93 |
| Branching points (#) | 23.14 | ± | 5.52 | 19.17 | ± | 4.62 | 21.09 | ± | 6.04 | 23.10 | ± | 4.51 |
| Total tubes (#) | 47.29 | ± | 12.57 | 44.67 | ± | 12.61 | 43.27 | ± | 13.91 | 46.60 | ± | 9.00 |
| Total tube length (px) | 15415.00 | ± | 582.38 | 12292.00 | ± | 1169.33 | 16522.86 | ± | 1239.79 | 17933.00 | ± | 1858.48 |
| Total loops (#) | 10.43 | ± | 1.81 | 7.00 | ± | 2.19 | 12.91 | ± | 1.81 | 14.45 | ± | 1.97 |
| Total loop area ($px^2$) | 503384 | ± | 476226 | 374678 | ± | 331769 | 335432 | ± | 229543 | 279728 | ± | 170898 |
| Total loop perimeter (px) | 3087.11 | ± | 2047.40 | 3738.14 | ± | 2591.60 | 2833.46 | ± | 1241.24 | 2459.92 | ± | 1054.61 |

concentrations, with a small tendency for consumption as conditioning time progresses. Pyruvate is initially provided by the media, but the maintenance of its concentration in the media suggest intrinsic production by the MSCs, through non-glucose dependent alternative synthesis pathways. Acetate levels, an intermediate product for the aerobic pathway of pyruvate catabolism, also remain fairly constant, as it is not expected to accumulate, but to continue its path through the citric acid cycle towards energy production. The build-up of choline and formate (related to the production of acetate) converge to this assumption.

Evidence of anaerobic metabolism is also observed throughout the conditioning period, providing accumulation of ethanol and lactate, both of which derived mainly from fermentation processes of the same energetic precursor pyruvate.

The observed accumulation of metabolites related to both bioenergetic processes supports the combination of both aerobic and anaerobic pathways for energy production by conditioning MSCs populations.

Other substrates dynamics are worth highlighting. GlutaMAX$^{TM}$ is a formulation whose hydrolysis results in the release of alanine and, ultimately, of L-glutamine, which is an essential nutrient in cell cultures for energy production as well as protein and nucleic acid synthesis [55]. This profile of consumption of GlutaMAX$^{TM}$ and build-up of its by-products is observable in both cellular populations. Little consumption is detected at 24 hours, but the process becomes very significant after 48 hours in culture. Additionally, UC-MSCs seem to entail on this metabolic pathway faster that DPSCs. The metabolic dynamics observed supports the survival and biosynthetic pathway endured by MSCs that, amongst other compounds, results in the production and/or release of bioactive molecules into their surroundings.

Through the evaluation of the bioactive factors content of the UC-MSCs and DPSCs secretome, the paracrine function of MSCs in therapeutic setups is substantiated. This topic has been granted greater attention in research when compared to the metabolomic composition of the MSCs secretome, and various cell sources have been researched besides the bone marrow standard [56–64], such as the adipose tissue [65–68], the dental pulp and other dental tissues [62, 69–72], the umbilical cord tissue [64, 73], cord blood [74], and other placental/ foetal tissues [64, 75]. These works report the identification of most of the angiogenic factors herein investigated, such as Ang-2 [64], Endothelin-1 [66, 71], FGF-2 [58–60, 64, 65, 70, 74], G-CSF [68], HB-EGF [74], HGF [60, 65, 66, 73, 74], IL-8 [58, 64, 69, 73, 75], PLGF [59] [64], VEGF [56–63, 65, 66, 69–71, 74], and TGF-β [61, 65] [64].

We identified a series of pro-proliferative/ anti-apoptotic (TGFβs, FGF-2, G-CSF, HGF and VEGFs) and chemotactic factors (RANTES, GRO, MCP and MDC), as well as relevant cytokines (IL-6 and IL-8). Some other factors are described in the literature, which we were not able to detect in significant amounts, such as GM-CSF [65]. PDGF is another angiogenic factor

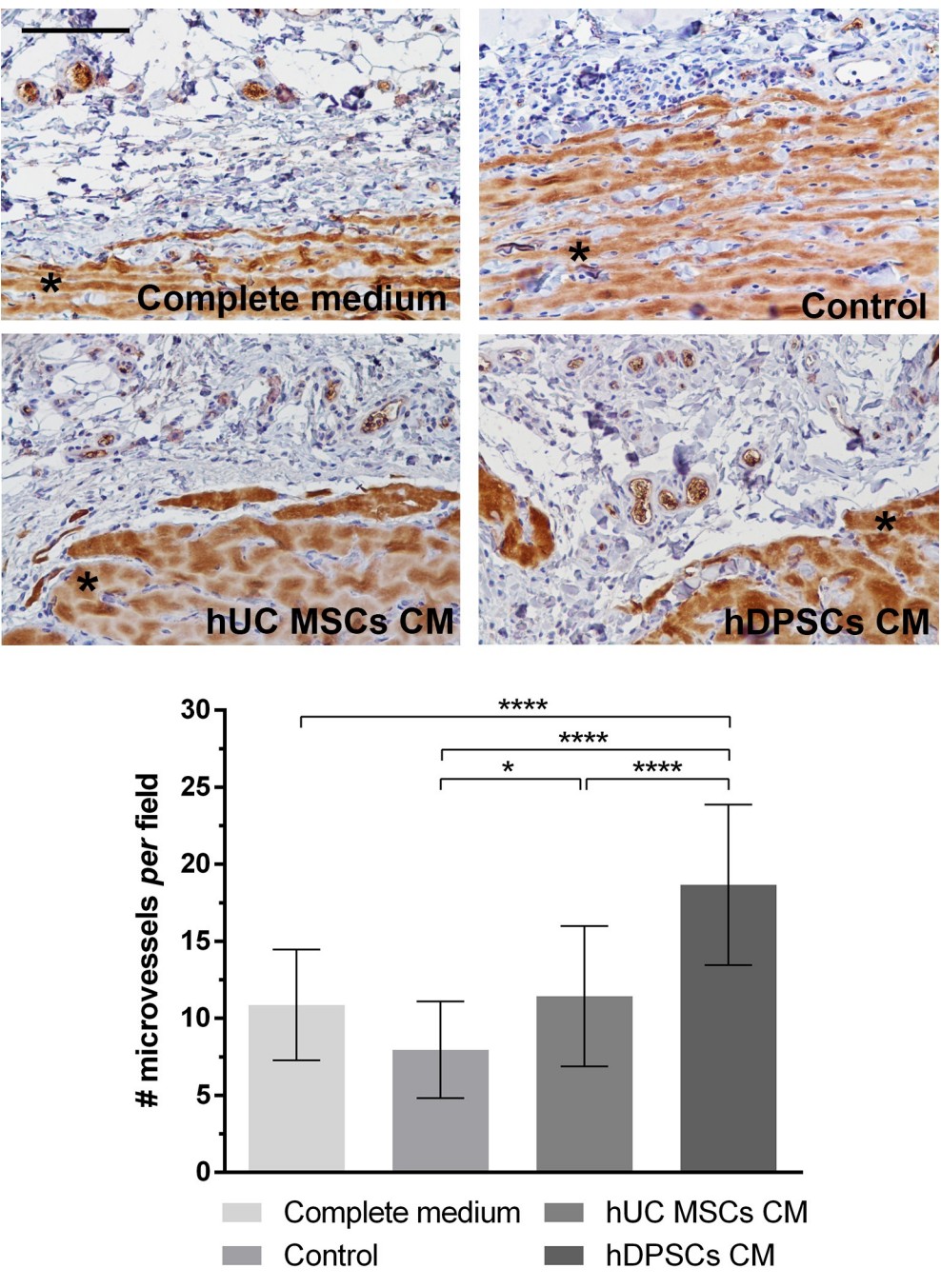

**Fig 9. *In vivo* re-vascularization (Matrigel® Plug) assay.** After 7 days of implantation of Complete, Control, UC-MSCs or DPSCs supplemented media groups. VEGFR2 immunohistochemical staining (upper panel); *: background uptake of DAB chromogen by the Matrigel®. Graphical representation of VEGFR2 immunopositive microvessels *per* field (#) observed in each group (lower panel) (scale bar = 100 μm). Values are presented as Mean ± SD. Significant differences indicated according to P values with one, two, three or four of the symbols (*) corresponding to 0.01≤P<0.05; 0.001≤P<0.01; 0.0001≤P<0.001 and P<0.0001, respectively.

previously reported to be produced by DPSCs [70] that we were unable to detect in the conditioned media.

**Table 11. *In vivo* re-vascularization (Matrigel® Plug) assay.** After 7 days of implantation of Complete, Control, UC-MSCs or DPSCs supplemented media groups. Microvessels *per* field at 200x magnification, following VEGFR2 immunohistochemical staining. Values are presented as Mean ± SD.

| *In vitro* re-vascularization after 7 days | Complete medium | | | Control | | | UCMSCs CM | | | DPSCs CM | | |
|---|---|---|---|---|---|---|---|---|---|---|---|---|
| **Microvessels *per* field (#)** | 10,87 | ± | 3,60 | 7,96 | ± | 3,14 | 11,43 | ± | 4,56 | 18,66 | ± | 5,21 |

Exploring into the specific functions of the identified factors, the VEGF family members are primary factors in the MSCs pro-angiogenic character [76, 77]. Interestingly, most authors do not take into account the existence of the different isoforms of VEGF, and rather quantify its presence in bulk. We herein demonstrate that the differential detection of VEGF isoforms is of great importance, since the assayed cellular sources demonstrated clear differences in the isoforms profile content. Human DPSCs were here demonstrated to predominantly secrete VEGF-A (that connects to endogenous re-vascularization response [78]), while UC-MSCs were more prone to VEGF-C secretion (prone to neurogenesis induction, without exerting angiogenic effects [79]), which may impact on their potential for tissue regeneration therapies. Janebodin and colleagues [62] also report source dependent differential isoform secretion, and demonstrated that DPSCs secreted superior levels of VEGF-A than BM-MSCs. They also report the secretion of VEGF-D, and to significantly greater extents than BM-MSCs, while we were unable to detect significant VEGF-D signal, especially after 48 hours of secretion. VEGFs (and Ang) directly stimulate endothelial cells populations to proliferate and migrate, and to organise in tube like vascular networks of increased stability and maturation [56]. Other mitogens are described for endothelial populations [80], such as FGF-2 and PLGF, which we failed to observe in relevant amounts [81], in line with the recent reports for UC-MSCs [64] and for DPSCs [63, 69]. HGF, the dominant bioactive factor identified in the UC-MSCs secretome, is recognised as anti-apoptotic and pro-mitogenic in various tissues [82], and was previously identified in UC-MSCs [77] and adipose derived-MSCs [65]. The strong expression of TGF-βs (particularly TGF-β1) by UC-MSCs could raise flags to their potential effect on the stimulation of fibroblastic population at a lesion site, but the combined secretion with antagonistic molecules, such as HGF and Follistatin may be the key to the antifibrotic character of MSCs populations [83]. IL-8 similarly displays pro-angiogenic features [65], promoting endothelial proliferation, migration and *in vitro* tube network formation [73].

These observations lead to the conclusion that both cell sources successfully supplied endothelial populations with stimulatory factors for their *in vitro* proliferation, migration and tube formation capacity, and forestalled senescence and apoptosis factors deprived endothelial cells. Indeed, the CM efficiently outweighed the complete endothelial cell expansion media utilised as positive control.

IL-8 and G-CSF are potent chemoattractant molecules [84], with recognised role in the homing of endogenous and delivered MSCs to lesion sites [85]. We observed great amounts of MCP-1 secretion by UC-MSCs, but not from DPSCs, colliding with Bronckaers' observations [69]. Potapova [58] reports BM-MSCs to also secrete MCP-1, in inferior proportion than IL-8 and IL-6. We observed an inverse pattern, since UC-MSCs secreted much expressive concentrations of MCP-1 instead of the ILs. Other entities such as eotaxin, fractalkine, GRO, and MDC and RANTES have been demonstrated to be chemoattract to MSCs [26, 86, 87]. The UC-MSCs maintain increased secretion of the molecules and, similarly to what was observed in the endothelial populations, the action of the CMs in the migration of the MSCs populations did not display equivalent differences. Conditioned media from both UC-MSCs and DPSCs promoted overall increased migration of the MSCs, seldom comparable to the stimuli provided by standard expansion media. The UC-derived populations appear to be more

responsive to the CM vehiculated stimuli. Additionally, the UC-MSCs also appear to respond to the bioactive factors produced by themselves during the migration challenge, as the un-supplemented controls display some degree of in-defect proliferation, exceeding that of DPSCs. The distinct response profile may relate to the expression of chemokine receptors by MSCs, that varies though species, culturing time and, suggestively, though tissue source for isolation [26].

Our report on the composition of the CMs is therefore consistent with the properties attributed to MSCs, suggesting that UC-MSCs provide a wider variety and greater concentration of relevant growth factors and cytokines. The proposed difference in the secretory potential of UC-MSCs and DPSCs was explored *in vitro* and *in vivo* focusing on their effects on endothelial populations, as well as on the MSCs themselves. The observed effects on the endothelial populations discarded the initial assumption that UC-MSCs presented increased potential than DPSCs, since both CM presented comparable stimulatory and protective effects on this population. Significant differences between the two media performances were seldom observed and, whenever present, indicated superior performance to DPSCs-CM supplemented groups. The variety of growth factors and observed effects emphasises on the complementarity of functions between the known and unknown factors that compose MSCs secretion cocktail [64]. This complementarity is also well patent in some works, where the abolishment of one factor only partially attenuate the effects displayed [59].

Amid the direct effects of the MSC's secretome on the angiogenic populations and, hence, on the revascularization of lesion sites, many of the bioactive factors herein identified exert additional effects on other cellular populations, which contribute to their its significantly broad therapeutic potential in multiple applications. The MSCs secretome has been described to positively interact with gastric mucosa epithelial cells [88], dermal fibroblasts [89, 90] and keratinocytes [90], pulmonary alveolar and small airway epithelial cells [91], retinal cells [92], hepatocytes [93], central and peripheral neuronal cells [94, 95], and osteoblasts [96], as well as on interacting inflammatory populations [96].

To the authors' knowledge, this is the first comparative approach to the umbilical cord and dental pulp derived stem/ stromal cells populations' metabolomic and bioactive secretion profiles. In summary, the present work provided insight on the metabolic profile of UC-MSCs and DPSCs in culture during the serum-free conditioning process through which MSCs conditioned media was obtained. Some differences were evidenced in the metabolite dynamics on the CM from MSCs derived from the umbilical cord stroma and from the dental pulp through the conditioning period (particularly on glucose metabolism), but similar global metabolic profiles are suggested. Limited literature is available to compare on the metabolic profiles of MSCs.

More prominent differences are highlighted for the bioactive factors content of these CMs, where FST, HGF, G-CSF, IL-8 and MCP-1 dominate in the UC-MSCs secretion, while VEGF-A and FST are the most prominently produced by DPSCs.

Finally, the distinct secretory cocktail did not result in significantly different effects on endothelial cell populations (proliferation, migration and tube formation capacity). The apparent decreased chemotactic factors content of DPSCs was additionally refuted by the comparable capacity to induce MSCs migration *in vitro*.

## Supporting information

**S1 Table. Statistically significant differences in Metabolites concentration (μM).** Calculated from 1H-NMR spectra of unconditioned (Plain Medium), and UC-MSCs and DPSCs conditioned media, after 24 and 48 hours. Significance of the results is indicated according to P

values with one, two, three or four of the symbols (*) corresponding to 0.01≤P<0.05; 0.001≤P< 0.01; 0.0001≤P<0.001 and P<0.0001, respectively; ns, not significant.
(DOCX)

**S2 Table. Statistically significant differences on detected Bioactive factors (pg/mL).** Calculated from Multiplexing LASER Bead Analysis of UC-MSCs and DPSCs Conditioned Media after 24 and 48 hours of conditioning. Significance of the results is indicated according to P values with one, two, three or four of the symbols (*) corresponding to 0.01≤P<0.05; 0.001≤P<0.01; 0.0001≤P<0.001 and P<0.0001, respectively; ns, not significant.
(DOCX)

**S3 Table. Significance of the results of the Presto Blue® Assay of UVECs.** At 24, 48, 72 and 96 hours of expansion in control or CMs supplemented media, according to P values with one, two, three or four of the symbols (*) corresponding to 0.01≤P<0.05; 0.001≤P<0.01; 0.0001≤P<0.001 and P<0.0001, respectively; ns, not significant.
(DOCX)

**S4 Table. Significance of the results of the Apoptosis (Annexin-V/ PI) assay of UVECs.** After 48 hours of expansion in Complete, Control or CMs supplemented media, according to P values with one, two, three or four of the symbols (*) corresponding to 0.01≤P<0.05; 0.001≤P<0.01; 0.0001≤P<0.001 and P<0.0001, respectively; ns, not significant.
(DOCX)

**S1 Dataset.**
(XLSX)

# Acknowledgments

The authors acknowledge MF Gärtner and laboratory technicians A Rema, from the Veterinary Pathology Laboratory–ICBAS, University of Porto, for their assistance in the histopathologic analysis of subcutaneous tissue. The author also thanks Professor Elísio Costa, from the Department of Biological Sciences of the Faculty of Pharmacy, University of Porto, for his insight on the bioactive factors data statistical treatment.

# Author Contributions

**Conceptualization:** Ana Rita Caseiro, Ana Colette Maurício.

**Data curation:** Ana Rita Caseiro, Sílvia Santos Pedrosa, Galya Ivanova, Mariana Vieira Branquinho, André Almeida, Irina Amorim, Tiago Pereira, Ana Colette Maurício.

**Formal analysis:** Ana Rita Caseiro, Sílvia Santos Pedrosa, Galya Ivanova, Mariana Vieira Branquinho, André Almeida, Irina Amorim, Tiago Pereira, Ana Colette Maurício.

**Funding acquisition:** Ana Colette Maurício.

**Investigation:** Ana Rita Caseiro, Sílvia Santos Pedrosa, Galya Ivanova, Mariana Vieira Branquinho, André Almeida, Fátima Faria, Irina Amorim, Tiago Pereira, Ana Colette Maurício.

**Methodology:** Ana Rita Caseiro, Sílvia Santos Pedrosa, Galya Ivanova, Mariana Vieira Branquinho, André Almeida, Fátima Faria, Irina Amorim, Tiago Pereira, Ana Colette Maurício.

**Software:** Sílvia Santos Pedrosa, Galya Ivanova, André Almeida.

**Supervision:** Ana Colette Maurício.

**Validation:** Ana Colette Maurício.

**Writing – original draft:** Ana Rita Caseiro, Sílvia Santos Pedrosa, Mariana Vieira Branquinho, André Almeida, Irina Amorim, Ana Colette Maurício.

**Writing – review & editing:** Ana Colette Maurício.

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
