## [Decision Letter · Decision Letter 0]

28 Aug 2019

PONE-D-19-21609

Mesenchymal Stem/ Stromal Cells metabolomic and bioactive factors profiles: a comparative analysis on the Umbilical Cord and Dental Pulp derived Stem/ Stromal Cells secretome

PLOS ONE

Dear Prof. Maurício,

Thank you for submitting your manuscript to PLOS ONE. After careful consideration, we feel that it has merit but does not fully meet PLOS ONE’s publication criteria as it currently stands. Therefore, we invite you to submit a revised version of the manuscript that addresses the points raised during the review process.

Although of some interest the manuscript has been found to be amended in several parts including figures.

The Authors in their revised manuscript must answer to all the requests of the two referees and highlight in red the alterations.

We would appreciate receiving your revised manuscript by Oct 12 2019 11:59PM. To enhance the reproducibility of your results, we recommend that if applicable you deposit your laboratory protocols in protocols.io, where a protocol can be assigned its own identifier (DOI) such that it can be cited independently in the future. For instructions see: http://journals.plos.org/plosone/s/submission-guidelines#loc-laboratory-protocols

We look forward to receiving your revised manuscript.

Kind regards,

Gianpaolo Papaccio, M.D., Ph.D.

Academic Editor

PLOS ONE

Journal Requirements:

NO

a) Please provide an amended Funding Statement that declares *all* the funding or sources of support received during this specific study (whether external or internal to your organization) as detailed online in our guide for authors at http://journals.plos.org/plosone/s/submit-now.  

b) Please state what role the funders took in the study.  If any authors received a salary from any of your funders, please state which authors and which funder. If the funders had no role, please state: "The funders had no role in study design, data collection and analysis, decision to publish, or preparation of the manuscript."

Reviewers' comments:

Reviewer's Responses to Questions

**Comments to the Author**

1. Is the manuscript technically sound, and do the data support the conclusions?

Reviewer #1: Yes

Reviewer #2: Yes

2. Has the statistical analysis been performed appropriately and rigorously? 

Reviewer #1: I Don't Know

Reviewer #2: Yes

3. Have the authors made all data underlying the findings in their manuscript fully available?

Reviewer #1: Yes

Reviewer #2: Yes

4. Is the manuscript presented in an intelligible fashion and written in standard English?

Reviewer #1: Yes

Reviewer #2: Yes

5. Review Comments to the Author

Reviewer #1: In this study, the authors identified several growth factors, cytokines and metabolites that are secreted by mesenchymal cells derived from dental pulp (DP-MSCs) and the umbilical cord (UC-MSCs) using Nuclear Magnetic Resonance Spectroscopy (NMR) and Multiplexing Laser Bear Analysis. They also determined the effect of DP-MSCs and UC-MSCs secreted factors on endothelial cells apoptosis, senescence, migration and vascularisation property. The identified secretomes provide maps of key factors that are involved in the role of MSCs in the regulation of surrounding microenvironment (particularly on angiogenesis) and therefore could have an impact in regenerative medicine.

Minor comments

The authors should discuss the potential effect of MSCs secreted factors on other cell types.

There are several grammatical errors which need to be corrected

Statistical (p value) should be added where required in the figure legends

Reviewer #2: In this paper authors conducted an analytical study focusing on the investigation of the metabolomic and bioactive factors composition of the secretome of hMSCs originated from two of the most promising sources for medical applications: the UC-MSCs and the DPSCs.

The study is interesting and well conducted.

Authors explained in details all the experiments done. Some corrections are needed.

Introduction should be reduced; moreover, to date, recent studies on DPSCs and UC-MSCs abilities are present (Cells. 2019 Mar 5;8(3);Stem Cell Research & Therapy vol 9, Art. num.: 236 (2018)).

In fig.3 Von Kossa staining is not clear; in my opinion, it should be better to present alizarin red staining also for qualitative evaluation. Moreover, authors should add scalbar to the pictures (Fig. 3; Fig. 7; Fig. 8; Fig. 9). In in vivo experiments, authors should confirm the vessel formation also by IF or IHC staining.

6. PLOS authors have the option to publish the peer review history of their article (what does this mean?). If published, this will include your full peer review and any attached files.

Reviewer #1: No

Reviewer #2: No

---

## [Author Response · Author response to Decision Letter 0]

11 Oct 2019

Academic Editor - Journal Requirements:

The Revised Manuscript addresses the PLOS ONE’s style requirements as defined in the aforementioned templates, particularly regarding Headings, Subheadings and Figure and Tables titles and legends.

2. PLOS ONE now requires that authors provide the original uncropped and unadjusted images underlying all blot or gel results reported in a submission’s figures or Supporting Information files. This policy and the journal’s other requirements for blot/gel reporting and figure preparation are described in detail at https://journals.plos.org/plosone/s/figures#loc-blot-and-gel-reporting-requirements and https://journals.plos.org/plosone/s/figures#loc-preparing-figures-from-image-files . When you submit your revised manuscript, please ensure that your figures adhere fully to these guidelines and provide the original underlying images for all blot or gel data reported in your submission. See the following link for instructions on providing the original image data: https://journals.plos.org/plosone/s/figures#loc-original-images-for-blots-and-gels .

No blot or gel figures are included in the Revised Manuscript.

NO

a) Please provide an amended Funding Statement that declares *all* the funding or sources of support received during this specific study (whether external or internal to your organization) as detailed online in our guide for authors at http://journals.plos.org/plosone/s/submit-now. 

This research was supported by Programa Operacional Regional do Norte (ON.2 – O Novo Norte), QREN, FEDER with the project “iBone Therapies: Terapias inovadoras para a regeneração óssea”, ref. NORTE-01-0247-FEDER-003262, and by the program COMPETE – Programa Operacional Factores de Competitividade, Projects PEst-OE/AGR/UI0211/2011 and PEst-C/EME/UI0285/2013 funding from FCT. This research was also supported by Programa Operacional Competitividade e Internacionalização (P2020), Fundos Europeus Estruturais e de Investimento (FEEI) and FCT with the project “BioMate – A novel bio-manufacturing system to produce bioactive scaffolds for tissue engineering” with reference PTDC/EMS-SIS/7032/2014 and by COMPETE 2020, from ANI – Projetos ID&T Empresas em Copromoção, Programas Operacionais POCI, by the project “insitu.Biomas - Reinvent biomanufacturing systems by using an usability approach for in situ clinic temporary implants fabrication” with the reference POCI-01-0247-FEDER-017771. 

This work received further financial support from the framework of QREN through Project NORTE-07-0124-FEDER-000066. The Bruker Avance III 600 HD spectrometer was purchased under the framework of QREN, through Project NORTE-07-0162-FEDER-000048 and is part of the Portuguese NMR Network created with support of FCT through Contract REDE/1517/RMN/2005, with funds from POCI 2010 (FEDER).

Ana Rita Caseiro (SFRH/BD/101174/2014) acknowledges FCT, for financial support. The authors acknowledge MF Gärtner and laboratory technicians A Rema, from the Veterinary Pathology Laboratory – ICBAS, University of Porto, for their assistance in the histopathologic analysis of subcutaneous tissue. The author also thanks Professor Elísio Costa, from the Department of Biological Sciences of the Faculty of Pharmacy, University of Porto, for his insight on the bioactive factors data statistical treatment.

b) Please state what role the funders took in the study. If any authors received a salary from any of your funders, please state which authors and which funder. If the funders had no role, please state: "The funders had no role in study design, data collection and analysis, decision to publish, or preparation of the manuscript."

(…..)

All figures were run through PACE diagnostic tool and were compliant with PLOS specifications. 

Reviewer #1: In this study, the authors identified several growth factors, cytokines and metabolites that are secreted by mesenchymal cells derived from dental pulp (DP-MSCs) and the umbilical cord (UC-MSCs) using Nuclear Magnetic Resonance Spectroscopy (NMR)and Multiplexing Laser Bear Analysis. They also determined the effect of DP-MSCs and UC-MSCs secreted factors on endothelial cells apoptosis, senescence, migration and vascularisation property. The identified secretomes provide maps of key factors that are involved in the role of MSCs in the regulation of surrounding microenvironment (particularly on angiogenesis) and therefore could have an impact in regenerative medicine.

Minor comments

1. The authors should discuss the potential effect of MSCs secreted factors on other cell types.

The authors included the following paragraph in section 3. Discussion and conclusions:

“Amid the direct effects of the MSC’s secretome on the angiogenic populations and, hence, on the revascularization of lesion sites, many of the bioactive factors herein identified exert additional effects on other cellular populations, which contribute to their its significantly broad therapeutic potential in multiple applications. The MSCs secretome has been described to positively interact with gastric mucosa epithelial cells [88], dermal fibroblasts [89, 90] and keratinocytes [90], pulmonary alveolar and small airway epithelial cells [91], retinal cells [92], hepatocytes [93], central and peripheral neuronal cells [94, 95], and osteoblasts [96], as well as on interacting inflammatory populations [96]. “

Therefore, the following references were included along the manuscript:

88. Xia X, Chiu PWY, Lam PK, Chin WC, Ng EKW, Lau JYW. Secretome from hypoxia-conditioned adipose-derived mesenchymal stem cells promotes the healing of gastric mucosal injury in a rodent model. Biochimica et Biophysica Acta (BBA)-Molecular Basis of Disease. 2018;1864(1):178-88.

89. Yoon BS, Moon J-H, Jun EK, Kim J, Maeng I, Kim JS, et al. Secretory profiles and wound healing effects of human amniotic fluid–derived mesenchymal stem cells. Stem cells and development. 2009;19(6):887-902.

90. Park S-R, Kim J-W, Jun H-S, Roh JY, Lee H-Y, Hong I-S. Stem cell secretome and its effect on cellular mechanisms relevant to wound healing. Molecular Therapy. 2018;26(2):606-17.

91. Akram KM, Samad S, Spiteri MA, Forsyth NR. Mesenchymal stem cells promote alveolar epithelial cell wound repair in vitro through distinct migratory and paracrine mechanisms. Respiratory Research. 2013;14(1):9. doi: 10.1186/1465-9921-14-9.

92. Johnson TV, DeKorver NW, Levasseur VA, Osborne A, Tassoni A, Lorber B, et al. Identification of retinal ganglion cell neuroprotection conferred by platelet-derived growth factor through analysis of the mesenchymal stem cell secretome. Brain. 2013;137(2):503-19.

93. Azhdari ZT, Mahmoodi M, Hajizadeh MR, Ezzatizadeh V, Baharvand H, Vosough M, et al. Conditioned Media Derived from Human Adipose Tissue Mesenchymal Stromal Cells Improves Primary Hepatocyte Maintenance. Cell journal. 2018;20(3):377-87.

94. Teixeira FG, Panchalingam KM, Assunção-Silva R, Serra SC, Mendes-Pinheiro B, Patrício P, et al. Modulation of the Mesenchymal Stem Cell Secretome Using Computer-Controlled Bioreactors: Impact on Neuronal Cell Proliferation, Survival and Differentiation. Scientific Reports. 2016;6:27791. doi: 10.1038/srep27791.

95. Assunção-Silva RC, Mendes-Pinheiro B, Patrício P, Behie LA, Teixeira FG, Pinto L, et al. Exploiting the impact of the secretome of MSCs isolated from different tissue sources on neuronal differentiation and axonal growth. Biochimie. 2018;155:83-91. doi: https://doi.org/10.1016/j.biochi.2018.07.026.

96. Li C, Li G, Liu M, Zhou T, Zhou H. Paracrine effect of inflammatory cytokine-activated bone marrow mesenchymal stem cells and its role in osteoblast function. Journal of Bioscience and Bioengineering. 2016;121(2):213-9. doi: https://doi.org/10.1016/j.jbiosc.2015.05.017. 

 2. There are several grammatical errors which need to be corrected.

The manuscript was revised for grammatical errors and correct in accordance. Further, Figures and Table numbers were updated.

 4. Statistical (p value) should be added where required in the figure legends. 

The p value detais were already presented in Fig 2 (former Fig 3) and 8 (former Fig 9) caption/ legend. Reference to the statistical analysis (significant differences and p values) information was included in the Fig 4 (former Fig 5) and 5 (former Fig 6) caption/ legends as recommended by the reviewer, as well as on the revised Fig 9 (former Fig 10).

Fig 2. Multilineage differentiation. (…) Results presented as Mean ± SEM. a: significantly different from undifferentiated group; b: UC-MSCs differentiated group significantly different from DPSCs differentiated group; c: UC-MSCs undifferentiated group significantly different from DPSCs undifferentiated group (P<0.05).

Fig 8. Tube formation assay. After 12 hours of UVECs exposure to Complete Control, UC-MSCs or DPSCs supplemented media (upper panel). Graphical representation of total branching points, total loops, covered area and total tube length observed in each group (lower panel)(scale bar = 200 μm). Values are presented as Mean ± SD. Significant differences indicated according to P values with one, two, three or four of the symbols (*) corresponding to 0.01≤P<0.05; 0.001≤P<0.01; 0.0001≤P<0.001 and P<0.0001, respectively.

Fig 4. Metabolite dynamics. Dynamics of GlutaMAXTM breakdown to glutamine and alanine of UC-MSCs (A) and DPSCs (B) populations; and Dynamics of Pyruvate catabolism into acetate, lactate and ethanol of UC-MSCs (C) and DPSCs (D) populations. Statistical significance of observed differences available in S1 Table, as Supporting Information.

Fig 5. Corrected absorbance readings of Presto Blue® Assay of UVECs. At 0, 24, 48, 72 and 96 hours of expansion in control or CMs supplemented media. Values are presented as Mean ± SD (A). Statistical significance of observed differences available in S3 Table, as Supporting Information; Apoptosis (Annexin-V/ PI) assay of UVECs, after 48 hours of expansion in Complete, Control or CMs supplemented media. Results presented as percentage (%) of cells of Viable, Early Apoptotic, Late Apoptotic and Death cells, as Mean ± SD (B). Statistical significance of observed differences available in S4 Table, as Supporting Information.

Fig 9. In vivo re-vascularization (Matrigel® Plug) assay. After 7 days of implantation of Complete, Control, UC-MSCs or DPSCs supplemented media groups. VEGFR2 immunohistochemical staining (upper panel). Graphical representation of VEGFR2 stained microvessels per field (#) observed in each group (lower panel)(scale bar = 100 μm). Values are presented as Mean ± SD. Significant differences indicated according to P values with one, two, three or four of the symbols (*) corresponding to 0.01≤P<0.05; 0.001≤P<0.01; 0.0001≤P<0.001 and P<0.0001, respectively.

-----------------

Reviewer #2: In this paper authors conducted an analytical study focusing on the investigation of the metabolomic and bioactive factors composition of the secretome of hMSCs originated from two of the most promising sources for medical applications: the UC-MSCsand the DPSCs. The study is interesting and well conducted. Authors explained in details all the experiments done. Some corrections are needed.

1. Introduction should be reduced; moreover, to date, recent studies on DPSCs and UC-MSCs abilities are present (Cells. 2019 Mar 5;8(3); Stem Cell Research & Therapy vol 9, Art. num.: 236 (2018)). 

The introduction section was revised and shortened according to the reviewer suggestions. 

Also, the recommended recent studies were considered in the reference list and the main conclusions contemplated in the Introduction section of the manuscript:

“Mesenchymal Stem/ Stromal Cells (MSCs) are at the forefront of research for the development of cell-based therapies, due to their capacity to self-renew and differentiate into several cell types, to secrete soluble factors with paracrine actions, as well as due to their immunosuppressive and immunomodulatory properties [1-6]. 

(…)

Currently, the umbilical cord stroma (Whärton jelly) and the dental pulp may come to gain ground as sources for MSCs-based therapies, due to the non/ minimally invasive and ethically accepted collection procedures (umbilical cords and extracted healthy teeth were previously considered medical waste), as well as for the increasingly available private and public banking options worldwide [12].

(…)

These observations were then attributed to the secretion products of those MSCs [18-20] and, in recent years, research has focused on deepening the knowledge on the effective composition of the MSCs secretion, in the form of soluble molecules or extracellular vesicles [6, 21-24].”

[6] Silachev DN, Goryunov KV, Shpilyuk MA, Beznoschenko OS, Morozova NY, Kraevaya EE, Popkov VA, Pevzner IB, Zorova LD, Evtushenko EA, Starodubtseva NL. Effect of MSCs and MSC-Derived Extracellular Vesicles on Human Blood Coagulation. Cells. 2019 Mar;8(3):258.

[12] Arutyunyan I, Fatkhudinov T, Sukhikh G. Umbilical cord tissue cryopreservation: a short review. Stem cell research & therapy. 2018 Dec;9(1):236.

2. In Fig.3 Von Kossa staining is not clear; in my opinion, it should be better to present alizarin red staining also for qualitative evaluation. 

The authors have substituted the Von Kossa staining for the Alizarin Red S, as suggested by the reviewer. Macroscopic images are presented, rather than microscopic record of the staining, due to the strong refingency resulting from the ARS uptake in the matrix, which precluded adequate fotographic record at higher magnifications. All neceessary sections were corrected accordingly :

Materials and Methods section:

2.1.3. Multilineage differentiation

Multilineage differentiation of the UC-MSCs and DPSCs was induced towards Osteogenic, Adipogenic and Chondrogenic phenotypes using specific differentiation media. Differentiation efficiency was assessed by Alizarin Red S, Oil Red O and Alcian Blue/ Sulfated Glycosaminoglycans (GAGs) quantification, as detailed in [42]. (…)

Results section :

Multilineage differentiation: The differentiation capacity of MSCs towards three mesodermal lineages was confirmed (Fig 3). Macroscopic observation revealed calcified matrix formation, evidenced by ARS staining in both UC-MSCs and DPSCs under osteodifferentiation conditions. (…)

Fig 3. Multilineage differentiation. A) Qualitative evaluation - Osteogenic differentiation: Alizarin Red S (ARS) staining after 21 days (scale bar = 6000 μm); Adipogenic differentiation: Oil Red O (ORO) staining after 14 days (scale bar = 100 μm); Chondrogenic differentiation: Alcian blue staining after 14 days (scale bar = 400 μm). B) Semi-Quantitative evaluation - Osteogenic differentiation: ARS concentration (μM) after 21 days; Adipogenic differentiation: Oil Red O (OD570nm) after 14 days; and C) Chondrogenic differentiation: Sulfated GAGs production (μg/ml) after 14 days, assessed by Blyscan™ Glycosaminoglycan Assay (Biocolor, UK). Control: Undifferentiated control; Results presented as Mean ± SEM. a: significantly different from undifferentiated group; b: UC-MSCs differentiated group significantly different from DPSCs differentiated group; c: UC-MSCs undifferentiated group significantly different from DPSCs undifferentiated group (P<0.05).

3. Moreover, authors should add scale bar to the pictures (Fig. 3; Fig. 7; Fig. 8; Fig. 9). 

The scale bars were included in Fig. 3, Fig. 7, Fig. 8, and Fig. 9 as suggested, and figure captions/ legend updated accordingly:

Fig 3. Multilineage differentiation. A) Qualitative evaluation - Osteogenic differentiation: Alizarin Red S (ARS) staining after 21 days (scale bar = 6000 μm); Adipogenic differentiation: Oil Red O (ORO) staining after 14 days (scale bar = 100 μm); Chondrogenic differentiation: Alcian blue staining after 14 days (scale bar = 400 μm).(…)

Fig 7. Migration assay of UVECs. At 0, 10, 14 and 16 hours in control or CMs supplemented media. Values are presented as percentage (%) of wound area coverage from the 0 hours baseline (scale bar = 200 μm). 

Fig 8. Migration assay of hMSC. At 0, 6 and 24 hours of expansion in control or CMs supplemented media. Values are presented as percentage (%) of wound area coverage from the 0 hours baseline (scale bar = 200 μm).

Fig 9. Tube formation assay. After 12 hours of UVECs exposure to Complete Control, UC-MSCs or DPSCs supplemented media (upper panel). Graphical representation of total branching points, total loops, covered area and total tube length observed in each group (lower panel)(scale bar = 200 μm).(…)

4. In in vivo experiments, authors should confirm the vessel formation also by IF or IHC staining.

The authors performed immunohistochemical staining of VEGF Receptor 2 expressing vascular sturctures in the Matrigel® plug containing tissues, to confirm the presence of local re-vascularization. The numbers of microvessels per field were further counted and adequate statistical treatment applied (by one-way ANOVA supplemented with Tukey’s HSD post-hoc test). All relevant details were added to the Materials and Methods section and complementary findings evidenced in the Results Section :

Materials and Methods section:

2.3.7. In vivo vascularization assay

(….) Collected tissues were processed for routine histopathologic analysis. Sequential sections (3 μm) were prepared and stained with haematoxylin-eosin (H&E) and assessed for the presence and pattern of endotheliocyte/ capillary penetration into the matrix plug. For the immunohistochemical study, sections were deparaffinized in xylene and rehydrated in sequential graded alcohols. Antigen retrieval was performed in EDTA, pH 8,0 for 30 minutes in water bath 100º C. The NovolinkTM Max-Polymer detection system (Novocastra, Newcastle, UK) was used for visualization, according to the manufacturer’s instructions. Slides were then incubated with anti-VEGFR2 antibody (clone 55B11; Cell Signaling Technology, Boston, USA), diluted 1:300, overnight at 4ºC. Colour was developed with 3.3- diamino-benzidine (DAB; Sigma, St. Louis, MO, USA) and sections were then counterstained with haematoxylin, dehydrated and mounted. Sections of normal rat skin were used as positive control and negative controls were performed by replacing the primary antibody with another of the same immunoglobulin isotype.

The tissue areas containing Matrigel® plugs and the surrounding tissue were microscopically evaluated in order to identify the regions of highest vascular density. Vessels which showed unequivocal brown VEGFR2 immunostaining were counted manually at higher magnification (×40) in at least 10 different regions of each sample.

Results section :

3.3.4. In vivo vascularization assay

After 7 days of subcutaneous implantation, the Matrigel® plugs were evidenced and collected. Discrete vascular penetration was observed macroscopically in all groups.

Microscopically, the Complete Medium and the UC-MSCs CM groups presented apparent increased capillary density than the Control group (Fig 10). In the CM groups, the capillary penetration reached deeper into the matrix. and displayed a more mature nature, evidenced by the patent vascular structures containing circulating erythrocytes These observations were further confirmed through VEGFR2 staining of the microvessels within the subcutaneous tissue at the penetrating interface with the Matrigel® plug (Fig 9 and Table 11). 

Table 11. In vivo re-vascularization (Matrigel® Plug) assay. After 7 days of implantation of Complete, Control, UC-MSCs or DPSCs supplemented media groups. Microvessels per field at 200x magnification, following VEGFR2 immunohistochemical staining. Values are presented as Mean ± SD.

In vitro re-vascularization 

after 7 days Complete medium Control UCMSCs CM DPSCs CM

Microvessels per field (#) 10,87 ± 3,60 7,96 ± 3,14 11,43 ± 4,56 18,66 ± 5,21

Fig 9. In vivo re-vascularization (Matrigel® Plug) assay. After 7 days of implantation of Complete, Control, UC-MSCs or DPSCs supplemented media groups. VEGFR2 immunohistochemical staining (upper panel); *: background uptake of DAB chromogen by the Matrigel®. Graphical representation of VEGFR2 immunopositive microvessels per field (#) observed in each group (lower panel)(scale bar = 100 μm). Values are presented as Mean ± SD. Significant differences indicated according to P values with one, two, three or four of the symbols (*) corresponding to 0.01≤P<0.05; 0.001≤P<0.01; 0.0001≤P<0.001 and P<0.0001, respectively.

---

## [Decision Letter · Decision Letter 1]

22 Oct 2019

Mesenchymal Stem/ Stromal Cells metabolomic and bioactive factors profiles: a comparative analysis on the Umbilical Cord and Dental Pulp derived Stem/ Stromal Cells secretome

PONE-D-19-21609R1

Dear Dr. Maurício,

We are pleased to inform you that your manuscript has been judged scientifically suitable for publication and will be formally accepted for publication once it complies with all outstanding technical requirements.

With kind regards,

Gianpaolo Papaccio, M.D., Ph.D.

Academic Editor

PLOS ONE

Additional Editor Comments (optional):

Reviewers' comments:

Reviewer's Responses to Questions

**Comments to the Author**

1. If the authors have adequately addressed your comments raised in a previous round of review and you feel that this manuscript is now acceptable for publication, you may indicate that here to bypass the “Comments to the Author” section, enter your conflict of interest statement in the “Confidential to Editor” section, and submit your "Accept" recommendation.

Reviewer #1: All comments have been addressed

Reviewer #2: All comments have been addressed

2. Is the manuscript technically sound, and do the data support the conclusions?

Reviewer #1: Yes

Reviewer #2: (No Response)

3. Has the statistical analysis been performed appropriately and rigorously? 

Reviewer #1: Yes

Reviewer #2: (No Response)

4. Have the authors made all data underlying the findings in their manuscript fully available?

Reviewer #1: Yes

Reviewer #2: (No Response)

5. Is the manuscript presented in an intelligible fashion and written in standard English?

Reviewer #1: Yes

Reviewer #2: (No Response)

6. Review Comments to the Author

Reviewer #1: The authors addressed all the reviewers' concerns and therefore, the reviewer has no further comments.

Reviewer #2: (No Response)

7. PLOS authors have the option to publish the peer review history of their article (what does this mean?). If published, this will include your full peer review and any attached files.

Reviewer #1: No

Reviewer #2: No

---

## [Editor Report · Acceptance letter]

18 Nov 2019

PONE-D-19-21609R1 

Mesenchymal Stem/ Stromal Cells metabolomic and bioactive factors profiles: a comparative analysis on the Umbilical Cord and Dental Pulp derived Stem/ Stromal Cells secretome 

Dear Dr. Maurício:

I am pleased to inform you that your manuscript has been deemed suitable for publication in PLOS ONE. Congratulations! Your manuscript is now with our production department. 

With kind regards,

on behalf of

Prof. Gianpaolo Papaccio 

Academic Editor

PLOS ONE